# A systematic evaluation of *Mycobacterium tuberculosis* Genome-Scale Metabolic Networks

**Víctor A. López-Agudelo**[1,2], **Tom A. Mendum**[3], **Emma Laing**[3], **HuiHai Wu**[3], **Andres Baena**[2,4], **Luis F. Barrera**[2,5], **Dany J.V. Beste**[3]*, **Rigoberto Rios-Estepa**[1]*

**1** Grupo de Bioprocesos, Departamento de Ingeniería Química, Universidad de Antioquia UdeA, Medellín, Colombia, **2** Grupo de Inmunología Celular e Inmunogenética (GICIG), Facultad de Medicina, Universidad de Antioquia UdeA, Medellín, Colombia, **3** Department of Microbial Sciences, Faculty of Health and Medical Sciences, University of Surrey, Guildford, United Kingdom, **4** Departamento de Microbiología y Parasitología, Facultad de Medicina, Universidad de Antioquia UdeA, Medellín, Colombia, **5** Instituto de Investigaciones Médicas, Facultad de Medicina, Universidad de Antioquia UdeA, Medellín, Colombia

* d.beste@surrey.ac.uk (DJVB); rigoberto.rios@udea.edu.co (RRE)

**Data Availability Statement:** All relevant data are within the manuscript and its Supporting Information files.

## Abstract

Metabolism underpins the pathogenic strategy of the causative agent of TB, *Mycobacterium tuberculosis* (Mtb), and therefore metabolic pathways have recently re-emerged as attractive drug targets. A powerful approach to study Mtb metabolism as a whole, rather than just individual enzymatic components, is to use a systems biology framework, such as a Genome-Scale Metabolic Network (GSMN) that allows the dynamic interactions of all the components of metabolism to be interrogated together. Several GSMNs networks have been constructed for Mtb and used to study the complex relationship between the Mtb genotype and its phenotype. However, the utility of this approach is hampered by the existence of multiple models, each with varying properties and performances. Here we systematically evaluate eight recently published metabolic models of Mtb-H37Rv to facilitate model choice. The best performing models, sMtb2018 and iEK1011, were refined and improved for use in future studies by the TB research community.

## Author summary

The tuberculosis bacillus, *Mycobacterium tuberculosis* (Mtb), is a global killer causing millions of deaths every year and is therefore a major burden to human health. Treatment of tuberculosis requires a cocktail of antibiotics for a minimum of 6 months. Treatment failure is common and is a major driver in the upward trend of antibiotic resistance, recognized by the World Health Organization as one of top ten threats to global health. A key to the success of Mtb as a human pathogen is ascribed to its extraordinary metabolic flexibility. Understanding the metabolism of Mtb is therefore an important goal of TB researchers as metabolic pathways present attractive drug targets. A powerful approach to study metabolism is through the use of genome-scale metabolic networks which enable metabolism to be studied at the whole system level rather than one enzyme at a time.

**Funding:** This work was supported by
COLCIENCIAS – Colombia, Grant 1115-5693-3520
(https://minciencias.gov.co/). DJVB received funds
(MR/K01224X/1) from Medical Research Council
(https://mrc.ukri.org/). VALA received funds from
COLCIENCIAS (National PhD scholarship Conv.
727-2015) (https://www.minciencias.gov.co/). The
funders had no role in study design, data collection
and analysis, decision to publish, or preparation of
the manuscript.

**Competing interests:** The authors have declared
that no competing interests exist.

Here, we comprehensively compare available genome scale metabolic networks. Our
results identify the best performing networks for a variety of modelling approaches. This
work allowed us to refine these models for the TB community to use in future studies to
probe the metabolism of this formidable human pathogen.

## Introduction

*Mycobacterium tuberculosis* (Mtb) is the causative bacterial agent of the global tuberculosis
(TB) epidemic, which is now the biggest infectious disease killer worldwide, causing 1.6 mil-
lion deaths in 2017 alone [1]. Mtb is an unusual bacterial pathogen, as it is able to cause both
acute life threatening disease and a clinically latent infections that can persist for the lifetime of
the human host [2,3]. Metabolic reprogramming in response to the host niche during both the
acute and the chronic phase of TB infections is a crucial determinant of virulence [4–7]. With
the worldwide spread of multi- and extensively-resistant strains of Mtb thwarting the control
of this global emergency, new drugs against Mtb are urgently needed and metabolic pathways
present attractive and potentially powerful targets [8,9].

Genome-scale constraint-based modelling has proved to be a powerful method to probe the
metabolism of Mtb. The first Genome Scale Metabolic Networks (GSMNs) of Mtb were pub-
lished in 2007 by Beste (GSMN-TB) [10] and Jamshidi (iNJ661) [10,11] and have been used as
a platform for interrogating high throughput 'omics' data, by simulating bacterial growth, gen-
erating hypothesis and informing drug discovery. Subsequently, these two original models
were iteratively improved to expand both their scope and accuracy [12–20], to give us a current
total of 16 inter-related GSMN of Mtb (Fig 1).

The first modifications to the two original models were carried out by Colijn *et al.* who
built MFF-RmwBo [21], by adding the mycolic acid producing sub-model of Raman *et al.*
(MAP, Fig 1) to GSMN-TB [22]. Fang *et al.* [23] systematically modified iNJ661 to produce
iNJ661v, a model designed to describe Mtb growing *in vivo.* [24,25]. Bordbar *et al.* expanded
the utility of the Mtb GSMNs by building the first integrated human macrophage–Mtb
genome-scale reconstruction, iAB-AMØ-1410-Mt-661 [26]. This host-pathogen model com-
bined the original iNJ661 with a cell-specific alveolar macrophage model derived from the first
human metabolic reconstruction Recon 1 [27]. A 2017 update of this model was subsequently
used to evaluate 'omics' data and predict substrate availability within TB infected macrophages
[28]. These advances were followed by a further complex series of updates and mergers to pro-
vide the wide selection of models we have today. Chindelevitch *et al.* used the algorithm, Meta-
Merge [12], to combine GSMN-TB and iNJ661 to improve the predictive value for high
throughput genome essentiality data, while Lofthouse *et al.* [14] published GSMN-TB 1.1, an
improved and extended version of GSMN-TB that successfully predicted sole nitrogen and
carbon substrate utilization patterns. In 2014, Vashisht *et al.* published a curated and updated
genome-scale model (iOSDD890) based on iNJ661, informed by a comprehensive manual re-
annotation of the Mtb genome [16]. However, this model lacked β-oxidation pathways, ren-
dering it unable to grow on fatty acids [20]. Also in 2014, Rienksma *et al.* combined three of
the previously published models [15] to construct a new model, sMtb, followed, in 2018, with
an improved version (sMtb 2018) designed for modelling Mtb metabolism inside macrophages
[29,30]. Finally, the first consolidated GSMN, iEK1011 was constructed using standardized
nomenclature of metabolites and reactions from the BiGG database [31,32]. These updates
and revisions, combined with the availability of omics data have provided the TB community

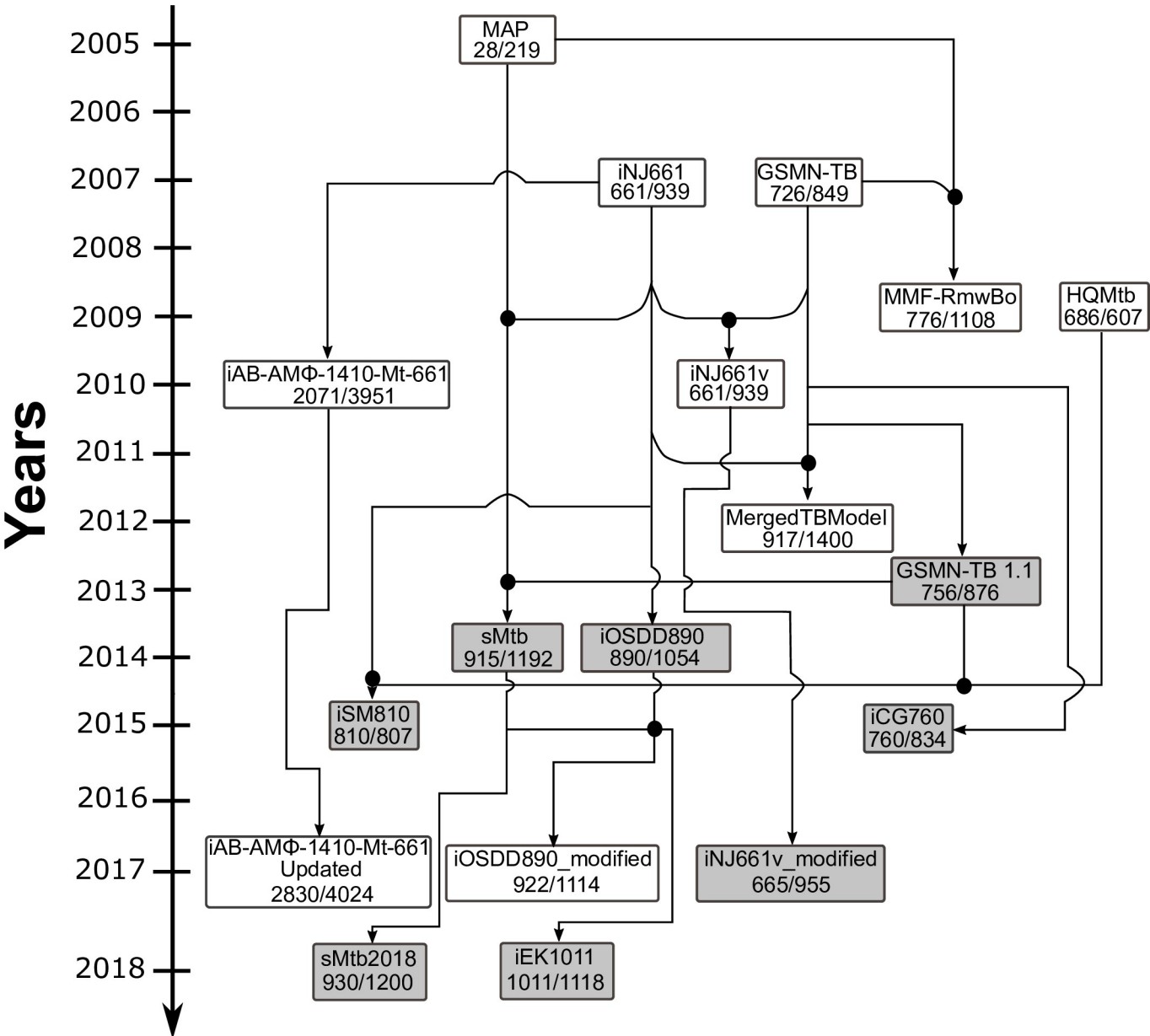

**Fig 1. The evolution of Genome Scale Metabolic models of Mtb.** Models highlighted in grey were analysed in this study. Numbers denote genes/intracellular reactions. Black circles are indicative of merged Mtb models.

with models that have better accuracy and scope when compared to earlier GSMN-TB iterations [33].

With so many well-annotated GSMN's of Mtb available (Fig 1), a crucial first step in any genome scale exploration of the metabolism of Mtb is the selection of an appropriate model. Here, we systematically evaluate the performance of eight recently published Mtb-H37Rv GSMNs. In addition to comparing the metrics of the models descriptively in terms of size, connectivity, number of blocked reactions and gaps in the network, we also identify the thermodynamically infeasible, and energy generating cycles that could significantly impact on the accuracy of flux simulations. Using Flux Balance Analysis (FBA) and Flux Variability Analysis

(FVA) we perform growth analysis and compare the models' ability to predict gene essentiality when grown on different carbon and nitrogen sources including cholesterol, a physiologically relevant carbon source for Mtb growing within its human host.

This work provides an inventory of the available GSMN-TB and their utility in recapitulating aspects of Mtb metabolism. In addition, we present updated versions of the best performing models iEK1011 and sMtb2018 (iEK1011_2.0 and sMtb2.0) for the TB research community to use in order to study the metabolism of this deadly pathogen.

## Results and discussion

### Descriptive evaluation of the models

Each of the GSMNs analysed in this study (Fig 1, S1 Appendix) combine knowledge from genome annotations, literature and measured biochemical compositions of Mtb. The complex linkage between genotype and phenotype is made by gene-protein-reaction (GPR) associations, implemented as Boolean rules in order to connect gene functions to enzyme complexes, isozymes or promiscuous enzymes, and finally to biochemical reactions [34]. Using set theory, we computed the intersection between all sets of the models' genes (Fig 2, and S1 Table). In accordance with expectations, the pairwise matrix (Fig 2) demonstrates that Mtb models constructed from the same ancestor (iNJ661 or GSMN-TB), are more similar (Fig 1, Fig 2). By contrast the consolidated models iEK1011 and sMtb2018 share gene similarities (>60%, <85% for iEK1011; and >60%, <98.4%) with all the other models demonstrating an independence from iNJ661 and GSMN-TB.

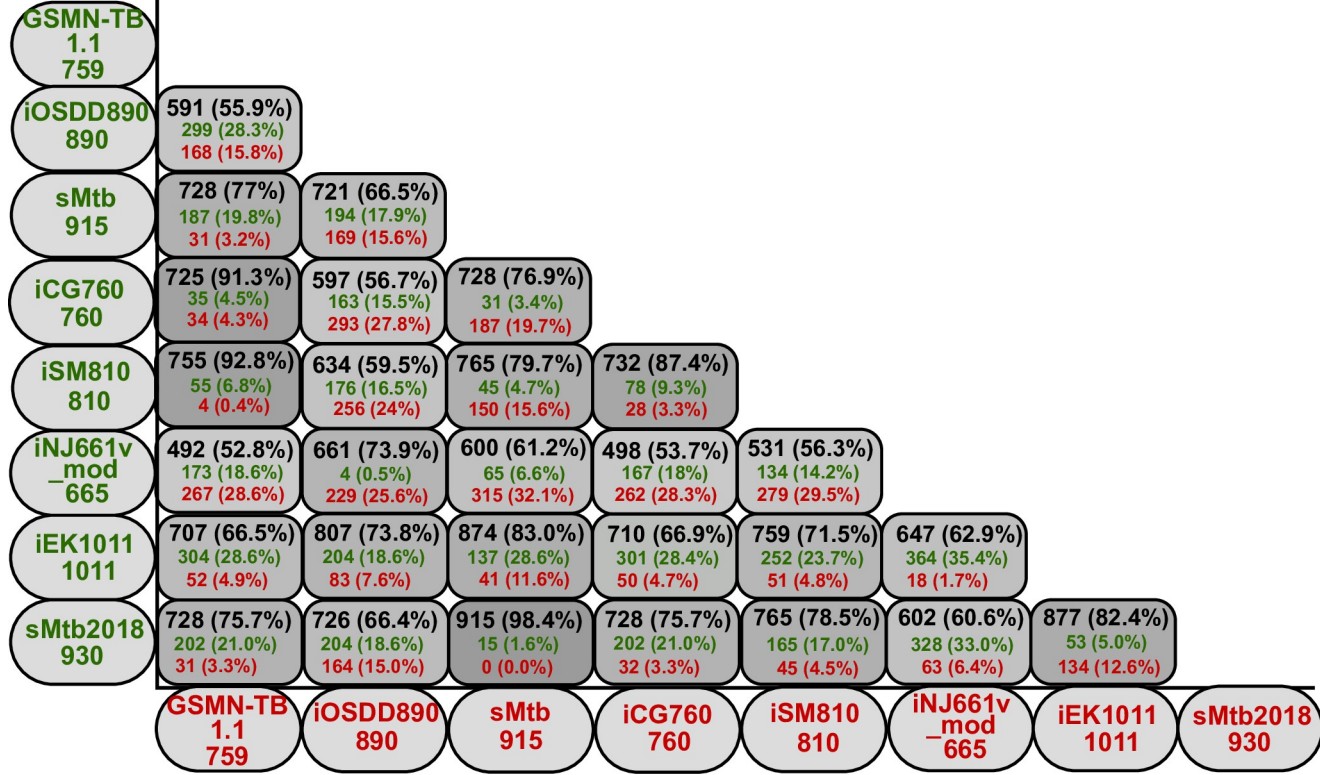

**Fig 2. Pairwise matrix of shared genes among Mtb models.** Values in black (genes in common between the models), green and red text represent the number and percentage of Mtb model genes specified in the y- and x-axis, respectively.

All the models contain essential metabolic pathways such as carbon, nitrogen, nucleotides, and cofactor metabolism (S2 Table), encoded by 479 common genes that can be used to construct a core metabolic network for Mtb [35]. The models sMtb, sMtb2018 and iEK1011 had the greatest coverage of GPR associations and contain genes associated with survival and virulence within the host such as transport, respiratory chain, fatty acid metabolism, dimycocerosate esters and mycobactin metabolism (S3 Table) and are therefore good candidates to study Mtb metabolism during intracellular growth [36,37].

In contrast, iOSDD890 (Fig 1) contains a high percentage of genes associated with nitrogen, propionate, pyrimidine, peptidoglycan, pyruvate and cofactor metabolism, but has a lower percentage of genes associated with glycerophospholipid metabolism, cholesterol degradation and fatty acid biosynthesis. Likewise, the iNJ661v_modified model (Fig 1) has a small number of genes involved in lipid-metabolism (e.g. β-oxidation, cholesterol degradation, fatty acid biosynthesis, lipid biosynthesis and mycolic acid biosynthesis). These models therefore have limitations for *in silico* simulation of Mtb growing on these physiologically relevant lipid sources and therefore also modelling *in vivo* growth.

## Checking mass and charge balances of biochemical reactions

Currency metabolites like water, protons, ATP, and cofactors like NADH, NADPH, $FADH_2$, CoA, etc. are ubiquitous and essential for metabolism. The addition of these cofactor metabolites in GSMNs, and in particular their inclusion in the biomass reaction, considerably improves phenotype predictions and is a hallmark of good quality reconstructions [19,38]. In order to check currency metabolites we converted the GSMNs into substance graphs (using a local script) where metabolites (nodes) are connected by edges (undirected and unweighted) if they appear in the same reaction [39] and computed node degrees (number of edges connected to the node) (Table 1 and S4 Table).

This analysis indicated that GSMN-TB 1.1, iCG760 and iSM810 models have the lowest number of currency metabolites (Table 1). Water and protons were the most underrepresented metabolites (low degree values), indicating that these models may not be correctly balanced. Some GSMNs e.g. iCG760, are functional even in the absence of any currency metabolites however this will negatively affect predictions. It is important that biochemical reactions are charge and mass balanced. Unbalanced reactions in GSMNs may allow proton or ATP

**Table 1. Degree values for currency metabolites of Mtb GSMNs.** PI: Phosphate, PPI: diphosphate, ACP: acyl-carrier protein, MK: menaquinone.

| Currency Metabolite | GSMN-TB1.1 | iOSDD890 | sMtb | iCG760 | iSM810 | iNJ661v_mod | iEK1011 | sMtb2018 |
|---|---|---|---|---|---|---|---|---|
| H | 72 | 701 | 512 | 102 | 127 | 667 | 741 | 511 |
| $CO_2$ | 150 | 202 | 266 | 146 | 160 | 193 | 204 | 268 |
| $H_2O$ | 5 | 507 | 624 | 0 | 5 | 472 | 569 | 624 |
| ATP | 236 | 353 | 320 | 264 | 242 | 295 | 315 | 320 |
| AMP | 101 | 170 | 115 | 103 | 103 | 117 | 132 | 115 |
| PI | 189 | 293 | 266 | 213 | 196 | 268 | 293 | 267 |
| PPI | 175 | 235 | 180 | 177 | 180 | 168 | 196 | 181 |
| COA | 175 | 163 | 230 | 175 | 190 | 142 | 194 | 230 |
| ACP | 98 | 48 | 136 | 94 | 103 | 48 | 57 | 136 |
| NADH | 102 | 141 | 262 | 104 | 113 | 111 | 176 | 263 |
| NADPH | 133 | 151 | 192 | 132 | 146 | 147 | 150 | 192 |
| $FADH_2$ | 40 | 47 | 55 | 42 | 48 | 21 | 62 | 55 |
| MK | 37 | 20 | 11 | 37 | 37 | 20 | 18 | 20 |
| $O_2$ | 37 | 76 | 64 | 40 | 41 | 68 | 91 | 65 |

**Table 2. Global features of the Mtb metabolic models analyzed in this study.** DEM: Dead-End Metabolite, URs: Unbounded Reactions, TICs: Thermodynamically Infeasible Cycles, diss. Flux: dissipation flux, MW: Molecular Weight, UD: Undetermined.

| | GSMN-TB 1.1 | iOSDD890 | sMtb | iCG760 | iSM810 | iNJ661v_mod | iEK1011 | sMtb2018 |
|---|---|---|---|---|---|---|---|---|
| **Reactions** | 876 | 1152 | 1311 | 965 | 938 | 1054 | 1228 | 1321 |
| **Intracellular Reactions** | 876 | 1055 | 1192 | 864 | 938 | 956 | 1118 | 1200 |
| **Metabolites** | 667 | 961 | 1047 | 754 | 724 | 840 | 998 | 1049 |
| **DEMs** | 25 | 162 | 33 | 84 | 53 | 90 | 110 | 34 |
| **Blocked Reactions** | 98 (11%) | 290 (25%) | 92 (7%) | 89 (9%) | 117 (12%) | 153 (14%) | 138 (11%) | 92 (7%) |
| **Metabolites without Formula** | 667 | 0 | 2 | 754 | 724 | 0 | 4 | 2 |
| **Unbalanced Reactions** | UD | 78 | 12 | UD | UD | 13 | 4 | 8 |
| **URs** | 27 (3%) | 52 (5%) | 70 (6%) | 45 (5%) | 33 (3.5%) | 67 (7%) | 20 (2%) | 75 (6%) |
| **TICs** | 8 | 17 | 16 | 8 | 8 | 23 | 4 | 17 |
| **ATP diss. Flux** | 0 | 0 | 0 | 0.33 | 0 | 0 | 0 | 0 |
| **GTP diss. Flux** | 0 | 0 | 0 | 0.33 | 0 | 0 | 0 | 0 |
| **CTP diss. Flux** | 0 | 0 | 0 | 0.33 | 0 | 0 | 0 | 0 |
| **UTP diss. Flux** | 0 | 0 | 0 | 0.33 | 0 | 0 | 0 | 0 |
| **Biomass MW** | UD | 1.0070 | 1.0126 | UD | UD | 1.0094 | 1.0937 | 1.0126 |

production out of nothing [40]. In order to test whether the Mtb GSMNs are mass and charge balanced, we used the COBRA Toolbox function "checkMassChargeBalance" [41]. Unfortunately, we were not able to perform this analysis for GSMN-TB1.1, iCG760 and iSM810 due to the lack of standard metabolite formulas in these models (Table 2). iEK1011 has the lowest number of unbalanced reactions (4) compared with sMtb2018 (8), sMtb (12), iNJ661v_modified (13) and iOSDD890 (78). The majority of the unbalanced reactions belonged to cell wall biosynthetic pathways, including arabinogalactan, peptidoglycan, and mycolic acid biosynthesis (S5 Table) reflecting the difficulties in rebuilding accurate metabolite formula for complex cell wall components.

## Biomass composition

The biomass formulations for the published Mtb GSMNs have been extensively described elsewhere [10,11,15]. The GSMN-TB and iNJ661 models and their respective descendants have different biomass compositions. The biomass composition for GSMN-TB was derived experimentally from chemostat cultures as well as estimated from the literature, whereas iNJ661 biomass was estimated only from literature [11,15]. As a result, there are significant differences in the amount of lipid (56% in GSMN-TB versus 25% in iNJ661) and nucleic acids (6% in GSMN-TB and 26% in iNJ661) in the biomass formulations (S6 Table). Moreover, in order to facilitate modelling of the metabolism of Mtb both *in vitro* and *in vivo*, GSMN-TB has two biomass formulations: "BIOMASS1", containing the complete macromolecular components of Mtb and "BIOMASSe", consisting of only the bacterial components essential for *in vitro* growth.

## Growth-associated maintenance and biomass reactions

The growth-associated maintenance (GAM) is the amount of energy required to replicate the cell whereas the non-growth-associated maintenance (NGAM) [15,32] is the amount of ATP required to maintain survival in the absence of growth. For Mtb these values have been estimated using data from other bacteria as experimental data is not available. The Mtb models originating from iNJ661 use a GAM of 60 mmol gDW$^{-1}$; those derived from GSMN-TB have a GAM of 47 mmol gDW$^{-1}$ whilst those originating from sMtb use a GAM of 57 mmol gDW$^{-1}$.

The values for NGAM are within the range of 0.1 and 3.15 mmol gDW$^{-1}$ h$^{-1}$ and therefore have negligible effects on gene essentiality predictions [32].

A comparison between biomass reactions across Mtb GSMNs (S1 File) showed that the biomass reaction "BiomassGrowthInVitro" from sMtb and sMtb2018 cannot be produced in 7H9 medium containing glycerol, Tween and OADC (S7 Table). We found this was because sMtb and sMtb2018 are unable to produce spermidine and S-Methyl-5-thio-alpha-D-ribose1-phosphate in these conditions.

Another potential error in GSMNs is the molecular weight of biomass, which should be defined as 1 g/mmol. Discrepancy in biomass weight can arise as a result of unbalanced reactions which will affect the reliability of flux predictions using FBA. This effect can be amplified when host-pathogen interactions are simulated by integrating host and pathogen metabolic models [42]. Using a systematic algorithm [42] we found deviations of less than 10% from 1 g/mmol in all the Mtb models tested (Table 2) (iOSDD890 (0.7%), iNJ661v_mod (0.9%), sMtb (1.2%), sMtb2018 (1.2%) and iEK1011 (9%)) demonstrating that these models are suitable for modelling the metabolism of Mtb within the host. The biomass of iEK1011 has the highest value because this model is a hybrid of sMtb and the iOSDD890 biomass reactions (S8 Table).

## Blocked reactions and dead-end metabolites

Identifying blocked reactions within a GSMN is important for identifying metabolic dead zones caused by dead-end metabolites (metabolites that are not consumed) [43–45]. Using the MC3 algorithm [46], we show that Mtb models derived from GSMN-TB (GSMN-TB1.1, iCG760, and iSM810) have a smaller number of blocked reactions in comparison with the iNJ661 derived models (iNJ661v_modified, and iOSDD890) (Table 2). sMtb and sMtb2018 have the lowest percentage (7%) of blocked reactions, in contrast to iOSDD890, which has the highest percentage (25%). All the Mtb GSMNs included blocked reactions in lipid, cofactor, sugar and amino acid metabolism; iOSDD890, iSM810 and iNJ661v_mod had blockages in important pathways such as glycolysis and redox metabolism (S9 Table). Most of the models excluding iCG760 and iSM810 contained gaps in the vitamin B12 biosynthesis pathway [47]. Specifically, we found that aqua(III) cobalamin and different cobalt-precorrins were not connected by reactions in most of the networks. This cofactor is necessary for activation of essential pathways such as nucleotide, propionate, and amino acids metabolism [47]. The existence of a functional B12 biosynthetic pathway is still under debate. A bona-fide transporter of vitamin B12 has been identified [48,49], however there remains no direct evidence that Mtb is able to scavenge vitamin B12 from its intracellular niche [49,50].

Of those models that contained a pathway for cholesterol degradation (GSMN-TB1.1, iCG760, iSM810, iEK1011, sMtb, and sMtb2018) the GSMN-TB1.1 cholesterol degrading pathway contained a number of dead end metabolites making this model unsuitable for exploring the metabolism of this important *in vivo* carbon source.

## Thermodynamic and energetic properties

Integrating thermodynamics data into GSMNs is extremely useful in order to check the feasibility of reactions and their directionality [51,52]. Although, Mtb GSMNs have been built from thermodynamics information, current Mtb GSMNs have never been checked for infeasible internal flux cycles. These are reactions that do not exchange metabolites with the surroundings and therefore violate the second law of thermodynamics [51,53,54]. A tractable way to identify reactions participating in these thermodynamically infeasible cycles (TICs) is to define the set of reactions required for an unbounded metabolic flux under finite or zero substrate uptake inputs. Using FVA the Unbounded Reactions (URs) can be identified as those reactions

with fluxes at the upper and/or lower bound constraints. Thus we identified the thermodynamically infeasible cycles (TIC) using a local script following methodologies based on FVA and the analysis of the null space of the stoichiometric matrices (S10 Table, S2 File) [52,55]. Using this approach we show that models descended from GSMN-TB (GSMN-TB1.1, iCG760, and iSM810) have a lower percentage of unbounded reactions as compared with iNJ661 ancestors (iSM810, iNJ661v_modified). Interestingly, the sMtb2018 model has an increased number of unbounded reactions as compared to the original sMtb (Table 2).

Fritzemeier and colleagues demonstrated that over 85% of genome-scale models that lack exhaustive manual curation contain Energy Generating Cycles (EGCs) [56,57]. These cyclic net fluxes are entirely independent of nutrient uptakes (exchange fluxes) and therefore have a substantial effect on the predictions of constraint-based analyses, as they basically generate energy out of nothing. Using FBA with zero nutrient uptake [57] but maximizing energy dissipation reactions for ATP, GTP, CTP and UTP we show that iCG760 is the only Mtb genome-scale model that contains EGCs (Table 2).

## Gene essentiality metrics

An effective and commonly employed predictive matrix for GSMNs is the ability to reproduce high throughput gene essentiality data [58]. Several high throughput transposon mutagenesis screens have been performed for Mtb [59–65] in different *in vitro* conditions. To compare our models we used a transposon insertion sequence dataset produced by Griffin *et al* [62]. In this study genes were identified that were essential for growth on cholesterol as compared with glycerol [62]. Cholesterol is an important intracellular source of carbon when Mtb is growing within its host and cholesterol metabolism has been highlighted as a potential drug target [66]. This data was not, however used to identify the genes required for growth on cholesterol only. We therefore reanalyzed the Griffin transposon sequencing data using the statistical Bayesian/Gumbel Method incorporated into the software TRANSIT [67], to identify genes required for growth on glycerol, or growth on cholesterol (S11 and S12 Tables). Only genes categorized as essential (ES) and non-essential (NE) were considered for this analysis.

We evaluated the overall predictive power of all the Mtb GSMNs versus a total of four high throughput gene essentiality datasets [62,64,65] by computing the Area Under the Curve (AUC) of the Receiver Operating Characteristic (ROC) (S1 Fig). The predictive power of the six GSMNs that contain the cholesterol degradation pathway (GSMN-TB1.1, iCG760, iSM810, sMtb, sMtb2018 and iEK1011) showed that for both cholesterol and glycerol minimal media the models derived from GSMN-TB [10] as a core metabolic network have better predictive capacities than those using iNJ661 [11] (S13 and S14 Tables). However the recently curated model iEK1011 had the highest predictive capability overall. The supremacy of iEK1011 was also confirmed by comparing the predictive power of the models using essentiality data obtained for Mtb grown in standard Middlebrook 7H9 media [64] (S1C Fig and S15 Table).

We also used the essentiality dataset generated by Minato and colleagues who identified conditionally essential Mtb genes using several *in vitro* conditions including a complex medium "MtbYM", which contains several carbon and nitrogen sources and also amino acids, nucleotide bases, cofactors, and other nutrients [65]. Overall the Mtb GSMN were less able to correctly predict essentiality (S1D Fig and S16 Table) in MtbYM as compared to other media (S1A–S1C Fig), probably because the biomass objective functions were reconstructed and validated using growth on standard Mtb media [10,11]. However these analyses demonstrated the ability of these models to accurately predict gene essentiality under new nutritional conditions.

We identified genes that all of the GSMN's were unable to correctly assign essentiality (S17 Table, Fig 3). Using a fixed threshold value of 5% of the maximum wild-type growth rate

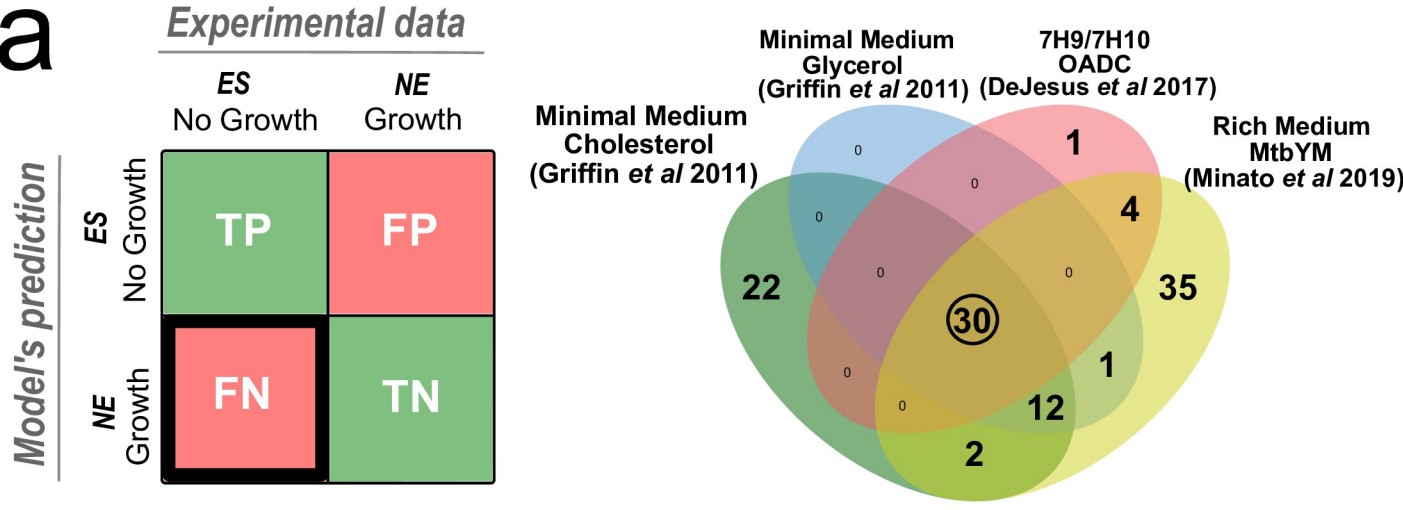

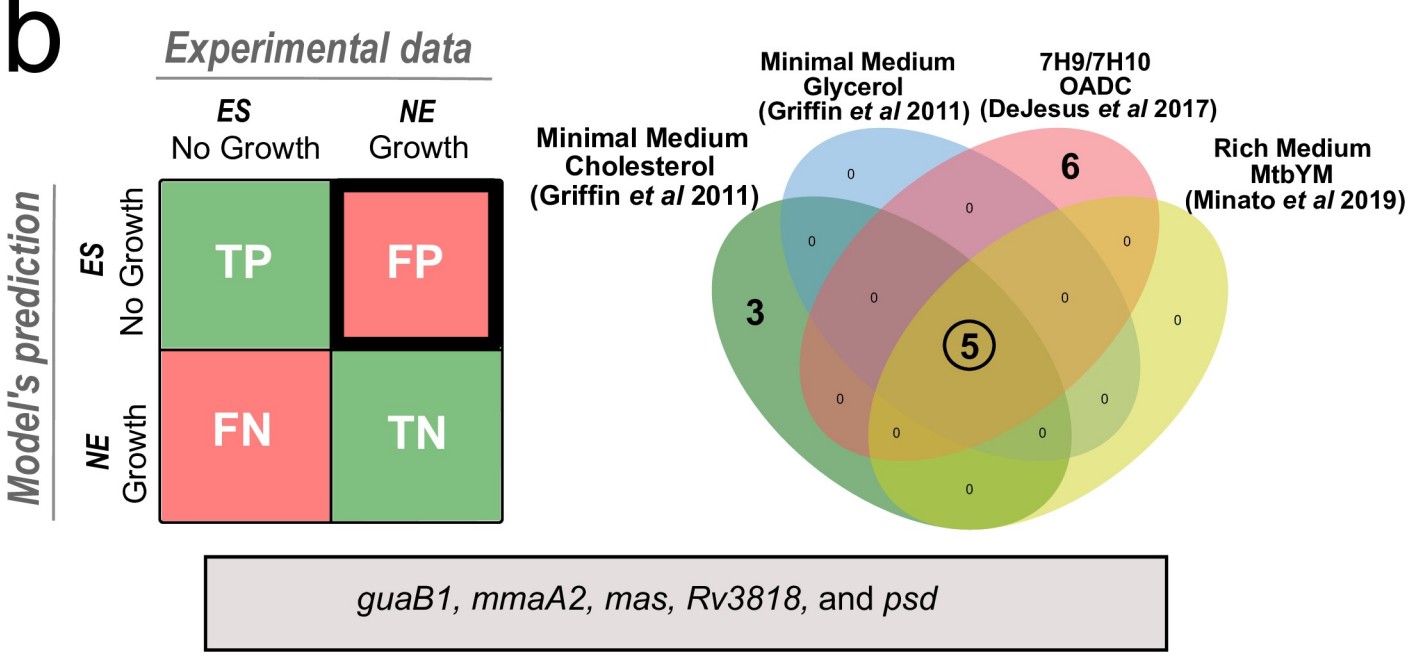

**Fig 3. False Negative and False Positive predictions in the evaluated media.** (a) Venn diagram for predicted false negative (FN) genes; (b) Venn diagram for predicted false positive (FP) genes. Genes in the grey box represents the intersection of all FN and FP genes in the four media. Genes are classified as True-positives (TP) if model simulation predicts no growth when essential genes are deleted, False-positives (FP) if model simulation predicts no growth when not essential genes are deleted, True negatives (TN) if model simulation predicts growth when not essential genes are deleted and False negatives (FN) if model simulation predicts growth when essential genes are deleted.

(WTGR) we identified *in silico* essential and non-essential genes and compared these to the experimental high throughput gene essentiality data to identify True Positives (TP), True Negatives (TN), False Positives (FP-a gene which is essential for *in silico* growth but non-essential

by Tn-seq) and False Negatives (FN-*in silico* the gene is non-essential but the biological data predicts essentiality). The FN included genes known to have a major role in Mtb central carbon metabolism e.g., *icl* (Rv0467, isocitrate lyase), *glt*A (Rv0896, Citrate synthase), *glp*D2 (Rv3302c, Glycerol-3-phosphate dehydrogenase), *pyk* (Rv1617, Pyruvate Kinase), *suc*C and *suc*D (Rv0951, Rv0952, Succinyl-CoA ligase), among others (S17A Table). Some of these genes e.g. *icl* and *glt*A are considered conditionally essential genes in the Online GEne Essentiality (OGEE) database [68], because they are classified as NE genes in 7H10 medium but ES in minimal medium [61,63]. This may reflect the presence of alternative routes *in silico* that are not feasible *in vivo* due to regulatory constraints. However, they may also reflect inaccuracies in the transposon mutagenesis studies. Some of the FP genes are involved in mycolic acid biosynthesis (S17B Table), e.g., *mma*A2 (Rv0644c, Cyclopropane mycolic acid synthase), and *mas* (Rv2940c, mycocerosic acid synthase). These genes were inaccurately classified as ES *in silico* but experimentally as NE. This reflects our incomplete knowledge of Mtb requirements for different mycolates and mycolate anabolism.

## Growth metrics

Mtb is able to metabolise several carbon and nitrogen sources both *in vitro* and when growing in the host [28,69–71], and therefore we evaluated the growth metrics of Mtb GSMNs on 30 sole carbon and 17 sole nitrogen sources (Fig 4). The *in silico* results were compared with available experimental data from Biolog Phenotype microarrays and minimal media [14,72]. Interestingly the recent consolidated models, iEK1011 and sMtb, had the poorest performance of all the models in predicting growth of Mtb in unique carbon and nitrogen sources (Fig 4A and 4B). A fundamental issue with the Mtb models descended from iNJ661 is that they all require glycerol for growth as this is a component of the biomass formulation. Both iEK1011 and sMtb were unable to grow *in silico* on cholesterol, acetate, oleate, palmitate and propionate when provided as sole carbon sources. We posit that this is a result of inaccuracies in reactions associated with redox metabolism and oxidative phosphorylation and specifically menaquinone-dependent reactions such as fumarate reductase and succinate dehydrogenase [73–76]. To test this hypothesis we added an irreversible menaquinone-dependent succinate dehydrogenase reaction into sMtb (Q[c] + SUCC[c] -> QH2[c] + FUM[c]). In support of our hypothesis this corrected the *in silico* growth phenotype of Mtb growing on acetate, cholesterol, propionate and fatty acids (Fig 4A and S18 Table). Although iEK1011 also contains a fumarate reductase reaction that is linked to menaquinone/demethylmenaquinone, it does not contain a menoquinone-dependent succinate dehydrogenase (the reverse reaction). As was the case for sMtb, the addition of a new irreversible menaquinone-dependent succinate dehydrogenase reaction (mqn8[c] + succ[c] -> fum[c] + mql8[c]) to iEK1011 significantly improves its growth predictions on sole carbon sources (S18I and S18J Table). These simulations are supported by experimental data demonstrating that fumarate reductase and succinate dehydrogenase are essential for Mtb to grow in media containing glycolytic and non-glycolytic substrates [74,75]. Succinate dehydrogenase is a bifunctional enzyme that is part of the TCA cycle and complex II of the electron transport chain, coupling the oxidation of succinate to fumarate, with the corresponding reduction of membrane-localized quinone electron carriers [75,77]. Mtb has multiple succinate dehydrogenases and fumarate reductases that are essential for the survival of Mtb during hypoxia [73–75,78,79]. Succinate is central to much of Mtb's lipid metabolism: host derived cholesterol, uneven chain length fatty acids or methyl branched amino acids all generate propionyl-CoA that can be channeled into the methylcitrate cycle to produce succinate (S2 Fig), while acetyl-CoA produced by β-oxidation of host derived even-chain fatty acids is metabolized through the glyoxylate shunt to also produce succinate (S2

## a

Mtb

| iNJ661v_mod | iOSDD890 | sMtb | GSMN-TB1.1 | iSM810 | iCG760 | iEK1011 | sMtb2018 | Carbon Source | EXP |
|---|---|---|---|---|---|---|---|---|---|
| *** | FN | FN | TP | TP | TP | FN | TP | Acetate | 1 |
| TP | TP | FN | TP | TP | TP | FN | TP | Oleate (9-Octadecenoate) | 1 |
| TP | TP | FN | TP | TP | TP | FN | TP | Palmitate(hexadecanoate) | 1 |
| TN | TN | TN | FP | FP | FP | TN | FP | L-Valine | 0 |
| TN | TN | FP | FP | FP | FP | TN | FP | L-Threonine | 0 |
| TP | TP | TP | TP | TP | TP | FN | TP | L-Serine | 1 |
| TP | TP | FN | TP | TP | TP | FN | TP | L-Proline | 1 |
| FP | FP | TN | TN | TN | TN | TN | TN | L-Phenylalanine | 0 |
| FP | FP | TN | TN | FP | FP | TN | FP | L-Ornithine | 0 |
| TN | TN | TN | TN | TN | TN | TN | TN | L-Methionine | 0 |
| TN | TN | TN | TN | TN | TN | TN | TN | L-Lysine | 0 |
| TN | TN | TN | TN | TN | TN | TN | TN | L-Leucine | 0 |
| FN | FN | FN | TP | TP | TP | FN | TP | L-Isoleucine | 1 |
| TN | TN | TN | TN | TN | TN | TN | TN | L-Histidine | 0 |
| TP | TP | FN | TP | TP | TP | FN | TP | L-Glutamate | 1 |
| TP | TP | TP | TP | TP | TP | FN | TP | Glycine | 1 |
| TP | TP | TP | TP | TP | TP | FN | TP | L-Aspartate | 1 |
| TP | TP | FN | FN | FN | FN | FN | FN | L-Arginine | 1 |
| TP | TP | TP | TP | TP | TP | FN | TP | L-Asparagine | 1 |
| TP | TP | TP | TP | TP | TP | FN | TP | L-Alanine | 1 |
| *** | *** | FN | FN | TP | TP | FN | TP | Cholesterol | 1 |
| TP | TP | FN | TP | TP | TP | FN | TP | Succinate | 1 |
| TP | TP | TP | TP | TP | TP | FN | TP | Pyruvate | 1 |
| TP | TP | FN | TP | TP | TP | FN | TP | Propanoate | 1 |
| TP | TP | TP | TP | TP | TP | FN | TP | L-Malate | 1 |
| TP | TP | TP | TP | TP | TP | FN | TP | L-Lactate | 1 |
| TP | TP | TP | TP | TP | TP | TP | TP | Glycerol | 1 |
| FP | FP | FP | FP | FP | FP | TN | FP | Glycerol-phosphate | 0 |
| FN | TP | TP | TP | TP | TP | TP | TP | Glucose | 1 |
| TP | TP | TP | TP | TP | TP | FN | TP | Citrate | 1 |

**MCC:** 0.46 | 0.52 | 0.38 | 0.59 | 0.58 | 0.58 | 0.17 | 0.58

## b

Mtb

| iNJ661v_mod | iOSDD890 | sMtb | GSMN-TB1.1 | iSM810 | iCG760 | iEK1011 | sMtb2018 | Nitrogen Source | EXP |
|---|---|---|---|---|---|---|---|---|---|
| TP | TP | FN | TP | TP | TP | FN | TP | L-Valine | 1 |
| TN | FP | FP | FP | FP | FP | FP | FP | L-Threonine | 0 |
| TP | TP | TP | TP | TP | TP | TP | TP | L-Serine | 1 |
| TP | TP | TP | TP | TP | TP | FN | TP | L-Proline | 1 |
| FP | FP | TN | TN | TN | TN | FP | TN | L-Phenylalanine | 0 |
| TP | TP | TP | FN | TP | TP | TP | TP | L-Ornithine | 1 |
| TN | TN | TN | TN | TN | TN | TN | TN | L-Methionine | 0 |
| TN | TN | FP | TN | TN | FP | TN | FP | L-Lysine | 0 |
| TN | TN | TN | TN | TN | TN | TN | TN | L-Leucine | 0 |
| FN | FN | FN | TP | TP | TP | FN | TP | L-Isoleucine | 1 |
| TN | TN | TN | TN | TN | TN | TN | TN | L-Histidine | 0 |
| TP | TP | TP | TP | TP | TP | TP | TP | L-Glutamate | 1 |
| TP | TP | TP | TP | TP | TP | TP | TP | Glycine | 1 |
| TP | TP | TP | TP | TP | TP | TP | TP | L-Aspartate | 1 |
| TP | TP | TP | FN | FN | TP | TP | TP | L-Arginine | 1 |
| TP | TP | TP | TP | TP | TP | TP | TP | L-Asparagine | 1 |
| TP | TP | TP | TP | TP | TP | TP | TP | L-Alanine | 1 |

**MCC:** 0.74 | 0.60 | 0.48 | 0.63 | 0.74 | 0.75 | 0.54 | 0.75

**Fig 4.** Predictive capacity of Mtb genome-scale models for the utilization of sole carbon and nitrogen sources; **(a)** Growth predictions of Mtb genome-scale models using sole carbon sources; **(b)** Growth predictions of Mtb genome-scale models by using sole nitrogen sources. Model's performance was evaluated by computation of the Matthews Correlation Coefficient (MCC). Experimental growth data were obtained from [14,72]. The values represent growth (value = 1) and no growth (value = 0) in a specific substrate, respectively. Carbon or Nitrogen substrates are classified as TP if the model predicts growth and growth is also observed experimentally, FP if the model predicts growth but no growth is observed experimentally, TN if the model and the experimental data both predict no growth and FN if model simulation predicts no growth but growth is observed experimentally.

Fig). Succinate oxidation by succinate dehydrogenases is therefore a critical step, as the enzyme couples the TCA cycle with electron transport chain and oxidative phosphorylation [78]. Having multiple succinate dehydrogenases provides Mtb with the metabolic flexibility to survive within the different niches within the human host.

Whilst carbon metabolism has been intensively studied *in vitro* and *ex vivo*, attention has only recently been directed to nitrogen metabolism [80–83]. Similar to carbon consumption, iEK1011 and sMtb were poor at predicting Mtb growth on sole nitrogen sources (Fig 4B, Matthews Correlation Coefficient (MCC) = 0.54 and 0.48, respectively). However, like carbon the addition of the menaquinone linked succinate dehydrogenase reaction into iEK1011 and sMtb significantly improves *in silico* growth on sole nitrogen sources (S19I and S19J Table). Specifically, correct growth predictions were obtained for Mtb growing on branched chain amino acids (isoleucine and valine) and proline (Fig 4B). This can be explained because complete degradation of these amino acids converges on succinate via methyl citrate cycle (degradation of isoleucine and valine) or the GABA shunt (degradation of proline) thereby coupling the TCA cycle with oxidative phosphorylation via succinate dehydrogenase.

## Refining Mtb GSMNs

Overall iEK1011 and sMtb2018 were the best GSMN's in terms of genetic background, network topology, number of blocked reactions, mass and charge balance reactions and gene essentiality predictions (Fig 2, Table 1, Table 2, and S1 Fig) and therefore we selected these models to refine further. iEK1011 has the advantage of containing standardized BiGG nomenclature of metabolites and therefore can easily be integrated into the human GSMN Recon3D [84] to simulate intracellular growth, while sMtb2018 has the utility that this model supports *in silico* growth in a wider variety of different nutritional conditions. Our analysis also highlighted some fundamental issues with these models which we analysed in order to improve the performance of these exemplar GSMN's.

As discussed above, including menaquinone and menaquinol as electron carriers in all respiratory chain reactions and selected ubiquinone-dependent reactions improved the GSMN's. Six new menaquinone-dependent reactions were added into the sMtb model e.g., succinate dehydrogenase, and cytochrome bc1 menaquinone-dependent, fumarate reductase, and malate dehydrogenase [30]. This improved the predictive growth metric of sMtb and importantly allowed *in silico* growth on cholesterol (see sMtb2018, Fig 4A). Similarly, we added to iEK1011 an irreversible menaquinone-dependent succinate dehydrogenase to improve the performance of this model when growing on media containing fatty acids and cholesterol. Further improvements were also made to cholesterol metabolism by updating both models to include reactions for the biochemical degradation of the C and D rings of cholesterol which was not known when these models were reconstructed (S20A and S20B Table) [85].

Although the functionality of a vitamin B12 biosynthetic pathway in Mtb remains uncertain, the detection of a small ratio of non-synonymous (dN) and synonymous (dS) nucleotide substitution (dN/dS < 1) in the cobalamin biosynthesis genes of clinical strains of Mtb suggests that this bacteria may be able to synthesize B12 in certain conditions [50]. Therefore,

until there is further experimental evidence to the contrary, we completed a B12 biosynthesis pathway by adding the genes Rv0306 and *cobCDU* to the models as well a B12 transporter (*Rv1819c and Rv1314c*), and added a dependence for B12 to MUTA (Methylmalonyl CoA Mutase) and METH (Methionine Synthase) reactions. We also included the co-factors biotin and pyridoxal-5-phosphate in the biomass formulation to enhance the phenotype prediction of sMtb2018 and iEK1011 as recommended by Xavier *et al.* [19].

Using the iNJ661v_modified model Xavier and colleagues demonstrated that inclusion of essential organic cofactors in biomass objective function improves phenotypic and gene essentiality predictions [19]. Here, the biomass reaction from iEK1011 was improved by the addition of universal cofactors such as sodium, NAD, NADP, CoA, FAD, FMN, Pyridoxal-5-phosphate, thiamine pyrophosphate, tetrahydrofolate, 5-formyltetrahydrofolate etc. to generate a new biomass formulation called "BIOMASS__2.1" (S20B Table). This biomass formula does not contain glycerol and therefore allowed this model, like Mtb itself, to grow in media lacking this carbon source. Similarly, we modified the biomass formula of sMtb to create "BiomassGrowth_2.0" which we incorporated into sMtb2.0.

We also added 51 missing genes into sMtb2018 that were identified from the iOSDD890 model and belong to pathways such as glycolysis, gluconeogenesis, TCA cycle, amino acid metabolism and mycolic acid biosynthesis to improve the GPR and predictive accuracy of this model (S21 Table).

We also identified (see method section) (S22 Table) [52,55] twelve TICs within sMtb2.0; (Fig 5A, S2 Appendix) and seven TICs in iEK1011_2.0 (Fig 5B, S3 Appendix); The major TIC of sMtb2.0 were within folate metabolism, catalysed by thymidylate synthase (thyA and thyX) and dihydrofolate reductase (DFRA). These reactions areessential steps for *de novo* glycine and purine biosynthesis and for the conversion of deoxyuridine monophosphate (dUMP) to deoxytimidine monophosphate (dTMP) (Fig 5A). DFRA 1 and DFRA 2, and DFRA 3 and DFRA4 are parallel reactions catalyzed by Rv2763c, dihydrofolate dehydrogenase. These reactions are identical except that they use a different currency metabolite (Fig 5A). Pereira and colleagues [38] recommend the use of NADPH/NADP in anabolic reactions and NADH/NAD$^+$ in catabolic reactions for more accurate flux distributions. Therefore we modified the model by retaining the DFRA2 and DFRA4 reactions and eliminating the NADH/NAD$^+$-dependent reactions, DFRA1 and DFRA3. Our thermodynamic calculations indicate that THYA and THYX ($\Delta_r G_{min}$ = -123 kJ/mol, $\Delta_r G_{max}$ = -9.3 kJ/mol and $\Delta_r G_{min}$ = -160 kJ/mol, $\Delta_r G_{max}$ = -33 kJ/mol, respectively) are irreversible in the forward direction (S23 Table) and therefore we also modified these reactions accordingly.

Our analysis also showed that two-ubiquinone oxidoreductases (QRr, NADH2r) and a transhydrogenase reaction (NADTRHD) were thermodynamically infeasible within the iEK1011_2 as both were identified as reversible. The Gibbs free energy computations indicate that these reactions are unidirectional in the direction of ubiquinol production ($\Delta_r G_{min} = -134.9 \frac{kJ}{mol}$, $\Delta_r G_{max} = -20.7 \ kJ/mol$ and $\Delta_r G_{min} = -131.7 \frac{kJ}{mol}$, $\Delta_r G_{max} = -20.1 \ kJ/mol$, respectively) and therefore we changed the model accordingly.

The modified model Mtb2.0 consists of 1322 reactions, 1054 metabolites and 989 genes, while iEK1011_2.0 comprises of 1238 reactions, 977 metabolites and 1012 genes. The predictive capability of these models was then evaluated by simulating gene essentiality predictions using available high-throughput essentiality experimental data [62,64,65] defining 5% of the wild-type growth rate as our arbitary essentiality threshold (S24 Table). This analysis showed that iEK1011_2.0 has the highest predictive performance in the four media conditions tested (glycerol and cholesterol minimal medium, Middlebrook 7H9, and YM medium) compared with all the Mtb GSMNs evaluated, including sMtb2.0 (Table 3). The ability of these updated

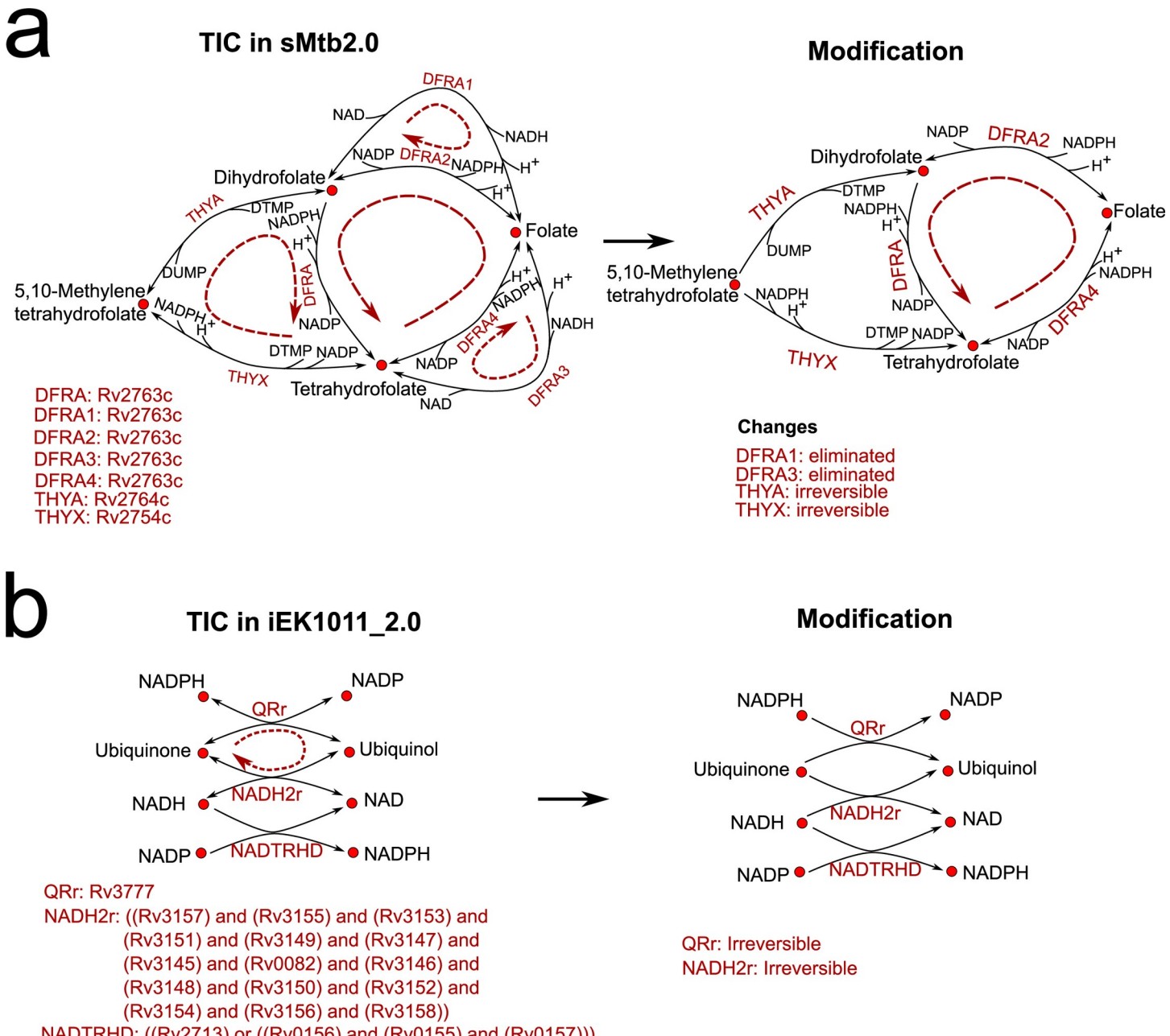

**Fig 5.** Thermodynamic Infeasible Cycles found in sMtb2.0 and iEK1011_2.0 with proposed modifications; **(a)** TIC at folate metabolism in sMtb2.0; **(b)** TIC affecting ubiquinone oxidoreductases in iEK1011_2.0.

models to predict growth on sole carbon and nitrogen sources was also improved (S25 Table). This included important carbon sources available in the human host and therefore iEK1011_2.0 and sMtb2.0 are suitable models for studying host-pathogen interactions [28].

The models iEK1011_2.0 and sMtb2.0 were then evaluated using MEMOTE [86], which is a standardised approach to quality control metabolic models. Overall scores for iEK1011_2.0 and sMtb2.0 were 74% and 37%, respectively (S3 File). The poor score for sMtb2.0 is misleading: it results from the lack of standardised nomencalature and does not reflect the model's

**Table 3. Genome-scale model features for sMtb2.0 and iEK1011_2.0.**

| Metrics | sMtb2.0 | iEK1011_2.0 |
|---|---|---|
| **Genes** | 989 | 1012 |
| **Reactions** | 1322 | 1238 |
| **Intracellular Reactions** | 1198 | 1119 |
| **Exchange Reactions** | 122 | 119 |
| **Metabolites** | 1054 | 977 |
| **% of Unbounded Reactions** | (49) 4% | 20 (1.8%) |
| **MCC Cholesterol minimal medium** | **0.55** | **0.63** |
| **Evaluated Genes** | 834 | 866 |
| True Positive Genes | 209 | 210 |
| True Negative Genes | 449 | 503 |
| False Positive Genes | 58 | 21 |
| False Negative Genes | 118 | 132 |
| **MCC Glycerol minimal medium** | **0.54** | **0.62** |
| **Evaluated Genes** | 834 | 866 |
| True Positive Genes | 207 | 206 |
| True Negative Genes | 449 | 503 |
| False Positive Genes | 58 | 21 |
| False Negative Genes | 120 | 136 |
| **MCC 7H9 medium** | **0.56** | **0.69** |
| **Evaluated Genes** | 984 | 1006 |
| True Positive Genes | 202 | 216 |
| True Negative Genes | 600 | 667 |
| False Positive Genes | 96 | 45 |
| False Negative Genes | 86 | 78 |
| **MCC MtbYM medium** | **0.43** | **0.52** |
| **Evaluated Genes** | 827 | 858 |
| True Positive Genes | 153 | 152 |
| True Negative Genes | 458 | 509 |
| False Positive Genes | 51 | 17 |
| False Negative Genes | 165 | 180 |
| **MCC Unique Carbon Source** | **0.58** | **0.67** |
| **Evaluated Metabolites** | 30 | 30 |
| True Positive Metabolites | 20 | 20 |
| True Negative Metabolites | 5 | 6 |
| False Positive Metabolites | 4 | 3 |
| False Negative Metabolites | 1 | 1 |
| **MCC Unique Nitrogen Source** | **0.75** | **0.75** |
| **Evaluated Metabolites** | 17 | 17 |
| True Positive Metabolites | 11 | 11 |
| True Negative Metabolites | 4 | 4 |
| False Positive Metabolites | 2 | 2 |
| False Negative Metabolites | 0 | 0 |

accuracy. Indeed the scores for the consistency category were 64% and 80%, for iEK1011_2.0 and sMtb2.0, respectively, demonstrating their high quality and utility in systems biology applications.

## Pathway utilization of sMtb2.0 and iEK1011_2.0

Using Roisin's Minimal Media containing glycerol and Tween80 (represented by oleic acid in the Mtb models) [70], we carried out Flux Variability Analysis (FVA) [87], FBA and uniform sampling using sMtb2.0 and iEK1011_2.0. FVA is a variant of FBA which, instead of finding a single optimal solution, computes the range of fluxes in each reaction that are compatible with optimization of the objective function [87,88]. A GAM value of 1 mmol gDW$^{-1}$ h$^{-1}$ and experimental uptake rates (glycerol, oleic acid and $CO_2$) from steady state chemostat cultures at a growth rate of 0.01 h$^{-1}$ [89] were used as constraints in both models. Complete results are reported in (S26A and S26B Table), but for brevity we discuss only the 33 reactions of Central Carbon Metabolism (CCM) and 21 extracellular (EX) reactions (S26C Table, S4 File) as informative examples. FBA using iEK1011_2.0 and sMtb2.0 predicts growth rates of 0.0084 h$^{-1}$ and 0.025 h$^{-1}$, respectively (S26C Table) showing that iEK1011_2.0 more accurately predicts experimental Mtb growth rate under these conditions [89].

Our FVA results showed that there were significant differences in the flux ranges when using the two models (p < 0.001; Kruskal-Wallis test). We hypothesized that this was a result of the different biomass formulations in the two models. In order to test this hypothesis we performed FVA without constraining the biomass objective function and in accordance with our expectations these simulations generated similar flux profiles (S27 Table, S4 File).

A comparison of the FVA results with the experimental $^{13}$C-Metabolic Flux profiles of chemostat grown Mtb indicates that the models are able to correctly predict the general experimental metabolic flux profile [89]. For instance, although sMtb2.0 has a higher flux distribution through gluconeogenic enzymes such as FBA and FBP compared to iEK1011_2.0 (Fig 6 and S26 Table), both values are comparable with the experimental flux values [89]. Flux through the non-oxidative enzymes of the pentose phosphate pathway enzymes, TKT and TAL, was greater in sMtb2.0 than in iEK1011_2.0 (Fig 6 and S26 Table) and the oxidative phase of the pentose phosphate pathway wasn't active in either of the models (Fig 6 and S26 Table).

Flux through the TCA cycle was slightly different between the two models however the general pattern was similar to the experimentally derived fluxes (S26 Table). sMtb2.0 predicted a lower carbon flux though the oxidative side of the TCA cycle than iEK1011_2.0 (Fig 6, S26 Table) and therefore was more aligned with the experimental data. Both models correctly predicted an active glyoxylate shunt and oxidation of pyruvate via the carbon fixing anaplerotic reaction, PCK, to produce oxaloacetate and succinyl-CoA through succinyl-CoA synthetase (Fig 6, and S26C Table). However, iEK1011_2.0 incorrectly predicts that this enzyme is functioning in the reverse direction producing succinate (S26C Table). Overall both models show utility in predicting experimental metabolic fluxes.

## Conclusions

By systematically evaluating eight of the recent Mtb GSMNs, we have highlighted the advantages and flaws of each of the models and identified solutions to some of their shortcomings. Importantly we have highlighted that the Mtb models descended from GSMN-TB (GSMN-TB1.1, iSM810 and iCG760) contain many unbalanced reactions, often because protons and water have not been accounted for. Dead-end metabolites, particularly in cofactor metabolism and related pathways was also an issue for some of the GSMNs. Overall, we show that sMtb2018 and iEK1011 have the best predictive power for Mtb. This analysis allowed us to update these two models by the addition of new reactions, gap filling of cofactor metabolism, and the identification and curation of TICs, to generate Mtb models with increased veracity.

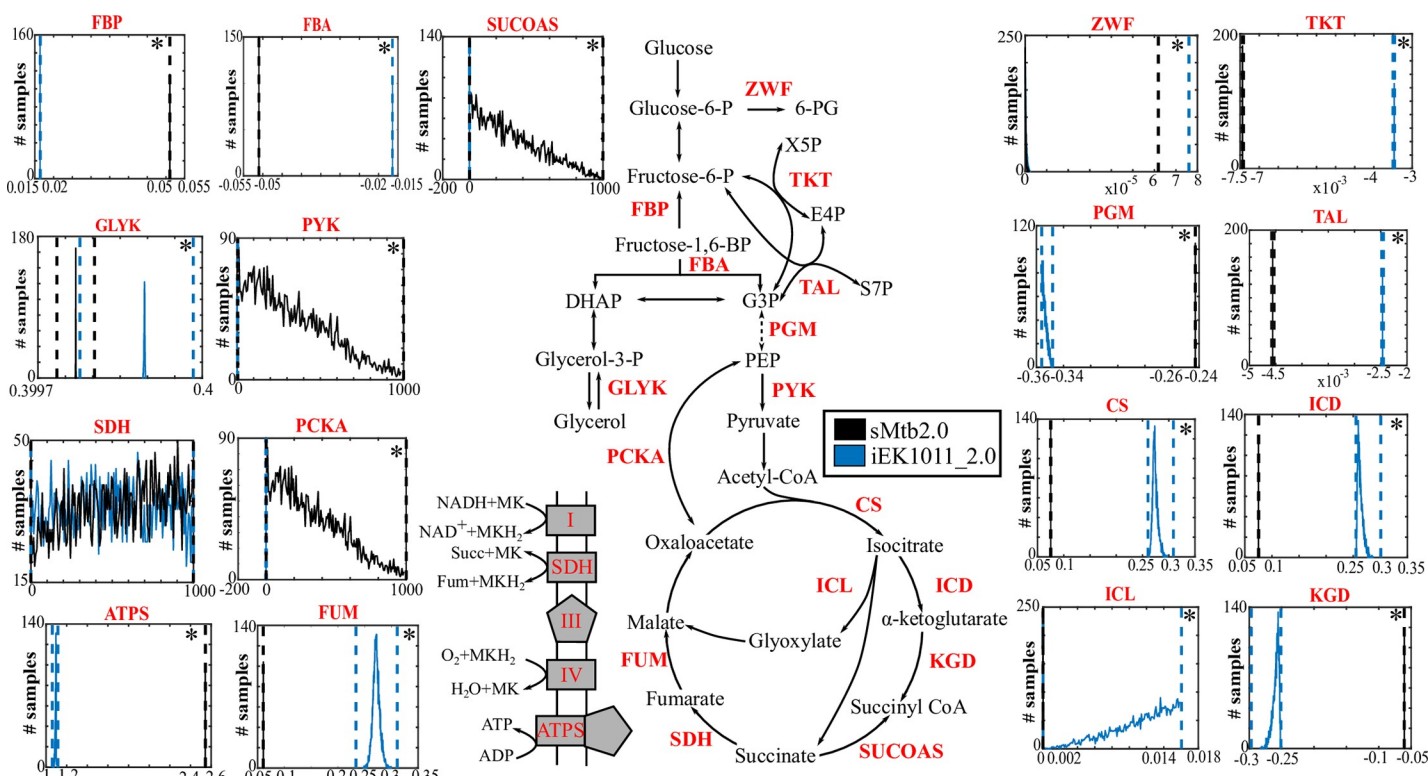

**Fig 6. Flux Sampling and FVA bounds of CCM reactions of sMtb2.0 and iEK1011_2.0 under Roisin's media and default biomass objective function.** The x-axis represents Flux values in mmol gDW$^{-1}$ h$^{-1}$. Dashed lines represent FVA bounds. Solid lines represent Flux sampling distributions. Reactions for which the two distributions are significantly different (p < 0.001; Kruskal–Wallis test) are marked with an asterisk in the top right corner. GLYK (Glycerol kinase), PGM (Phosphoglycerate mutase), PYK (Pyruvate kinase), PCKA (Phosphoenolpyruvate carboxykinase), CS (Citrate synthase), SDH (Succinate dehydrogenase), FUM (Fumarase), ATPS (ATP synthase), ZWF (Glucose 6-phosphate dehydrogenase), TKT (Transketolase), TAL (Transaldolase), ICL (Isocitrate Lyase), KGD (2-oxoglutarate dehydrogenase), FBA (Fructose-biphosphate aldolase), FBP (Fructose biphosphatase), ICD (Isocitrate dehydrogenase), SUCOAS (Succinyl CoA synthetase).

The improved GSMN's, sMtb2.0 and iEK1011_2.0, with their respective Memote report [86], are now available in sbml and json formats (S3 and S5 Files) to simulate and predict the metabolic adaptation of Mtb in a plethora of *in vitro* and *in vivo* intracellular conditions. We encourage researchers to continue to curate these models as new data and methods become available. Improved GSMN's including macrophage-Mtb models provide a critical platform for increasingly more accurate simulations and ultimately a better understanding of the underlying biology of this pathogen.

## Methods

All simulations were conducted on a laptop running Windows 10 (Microsoft) using MATLAB 2016a (MathWorks Corporation, Natick, Massachusetts, USA), COBRA Toolbox version 3.0 [41], RAVEN 2.0 [90] and Gurobi Optimizer version 7.5.2 (Gurobi Optimization, Inc., Houston, Texas, USA). All code written for this study is available in supplementary information (S1–S8 Files). Genome-scale models of Mtb Models were obtained from supplementary information of published papers and modified as follows:

- GSMN-TB1.1 –from [14] supplementary info.

- iOSDD890 –from [16] supplementary info.

- sMtb–from [15] supplementary info. Modification included were the addition of exchange reactions to allow constraints by growth medium components.

- iCG760 –from [17] supplementary info.

- iSM810 –from [18] supplementary info.

- iNJ661v_modified–from [19] supplementary info.

- sMtb2018 –from [30] supplementary info.

- iEK1011 –from [32] supplementary info.

### Network connectivity evaluation

GSMNs of Mtb were transformed to substrate networks by local scripts after eliminating bio-mass reaction. Node-specific topology metrics were carried out using the plugin Network Ana-lyzer [91] in Cytoscape 3.4 [92]. Two main topological parameters were evaluated for each model: 1) the node degree of each metabolite and 2) the clustering coefficient (S4 Table).

   MC3 Consistency Checker algorithm [46] was used to identify Single Connected and Dead End metabolites, and zero-flux reactions in each metabolic network model of Mtb. This algo-rithm uses a stoichiometric-based identification of metabolites connected only once in each metabolic network and utilize Flux-Variability-Analysis (FVA) for identifying reactions that cannot carry flux [87].

### Biomass Molecular Weight Check

Testing the biomass Molecular Weight consistency was done by running the script of Chan and colleagues [42]. A biomass reaction is not standardized when the Molecular Weight of the biomass formula is not equal to 1 g/mmol. However, the accuracy of the results relies on the correct chemical formulae of metabolites in the tested GSMNs. Furthermore, we check charge and mass balance of all Mtb GSMNs (S6 File).

### Identification of Unbounded Reactions (URs)

A straightforward way to identify all reactions that participate in one or more TICs is by per-forming flux variability analysis (FVA). All the Infeasible loops are evidenced as a set of reac-tions able to carry an unbounded metabolic flux under finite or even zero substrate uptake inputs. The URs are those reactions that by applying FVA [87], their fluxes will hit the values defined by the upper and/or lower bounds constraints [55]. Therefore, we performed FVA with the eight Mtb GSMNs with all the uptake media constraints defined by 1.0 mmol/gDW/h (S2 File).

### Identification of the core set of TICs

Schellenberger and colleagues [93] used a methodology for identifying the core set of TICs, which form the basis of all such possible cycles. This core set can be obtained by the computa-tion of the null space basis of the stoichiometric matrix (all possible thermodynamically infea-sible cycles form the null space of the stoichiometric matrix). Consequently, the set containing all the reactions that we previously identified participate in TICs was used to build a stoichio-metric matrix. Therefore, the null space basis of this set was computed and the different cycles composed by two and more reactions were identified by a local script (S2 File).

## Checking the existence of Energy Generating Cycles (EGCs)

Energy generating cycles (EGCs) exist in metabolic networks and can charge energy metabolites like ATP, GTP, CDP, and UTP without any input of nutrients; therefore, their elimination is essential for correcting energy metabolism [57,94]. Fritzemeier and colleagues developed a methodology for identifying if genome-scale models contain EGCs [57]. We applied it in two steps (S7 File): 1) Addition of Energy dissipation reactions (EDR) for ATP, GTP, CTP and UTP in the form: $H_2O[c] + XTP[c] \rightarrow H[c] + XDP[c] + Pi[c]$ and 2) maximization of each EDR flux $v_d$ while no substrate uptake is allowed into the model as follows:

$$\max v_d \tag{1}$$

$$s.t \quad Sv = 0 \tag{2}$$

$$\forall i \notin E : v_i^{LB} \leq v_i \leq v_i^{UB} \tag{3}$$

$$\forall i \in E : v_i = 0 \tag{4}$$

Here, $S$ is the stoichiometric matrix, $v$ the vector of fluxes, d the index of EDRs, $v_i^{LB}$ and $v_i^{UB}$ the vector of lower and upper bounds, respectively, and $E$ is the set of indices of all exchange reactions of the model. If the optimal value of $v_d$ for this optimization is $v_d>0$, there exist in the genome-scale model at least one EGC that is able to generate energy metabolites like ATP, GTP, CTP, or UTP.

## Curation of TICs

Two types of modifications were performed on the curated Mtb genome-scale metabolic network in order to eliminate TICs [55].

i. TICs formed by linearly dependent reversible reactions: Usually, these arise when there are two reactions ($NAD^+$- and $NADP^+$-dependent) with the same catalytic activity. In this instance, we forced the use of $NADPH/NADP^+$ in anabolic reactions and $NADH/NAD^+$ for catabolic reactions, as recommended by Pereira and colleague [38]. If two irreversible reactions that catalyze the forward and backward direction exist, both reactions (and GPR rules) are lumped together in just one reversible reaction.

ii. TICs formed by erroneous directionality assignments: we restricted the reaction directionality based on Gibbs free energy change (NExT algorithm) [95,96] as long as gene essentiality predictions are not compromised

The later modification was based on the utilization of the NExT (network-embedded thermodynamic analysis) algorithm [95,96]. This algorithm allows the identification of new irreversible reactions by calculating the thermodynamically feasible range of Gibbs energy of reactions and metabolite concentrations. NExT was implemented for those reactions participating in TICs under Matlab [96] with physiological conditions adapted for Mtb (Table 4). In the absence of intracellular metabolite concentration data we assumed that all metabolites are between 0.0001 mM and 10 mM, which represents a range of observed physiological concentrations used by Martinez *et al* [95,96].

Standard Gibbs energy of formation ($\Delta_f G_i$) (in kJ/mol), number of hydrogen atoms, and charge of all metabolites involved in TICs were obtained from the Biochemical Thermodynamic Calculator, eQuilibrator [104,105].

**Table 4. Biophysical properties and concentration ranges for intracellular Mtb.**

| Properties | Values | Reference |
|---|---|---|
| Redox potential, cytosol | -275 mV | Bhaskar et al., 2014 [97] |
| pH, intracellular | 5.7 | Zhang et al., 2003 [98] |
| pH, extracellular (activated macrophage) | 4.5 | Rohde et al., 2007; Vandal et al., 2009 [99,100] |
| Ionic strength | 0.15 M | Kümmel et al., 2006; Martínez et al., 2014 [96,101] |
| Oxygen Concentration (mM) | 0.0001–0.1 | Martínez et al., 2014 [102] |
| $[CO_2],[Pi]$ (mM) | 1–100 | Kümmel et al., 2006; Haraldsdóttir et al., 2012 [101,103] |
| Other metabolites (mM) | 0.0001–0.1 | Martínez et al., 2014 [96] |
| NADH/NAD | 0.0001–0.1 | Martínez et al., 2014 [96] |
| NADPH/NADP | 0.0001–0.1 | Martínez et al., 2014 [96] |

If a reaction was specified to be reversible in the set of TICs and had its maximum $\Delta_r G$ calculated to be negative, the reaction is considered to occur in the forward direction. In contrast, if the minimum $\Delta_r G$ was positive, the reaction is considered to occur in the reverse direction. No direction can be inferred when the minimum $\Delta_r G$ is negative and the maximum is positive. Changes in directionality of reactions were done strictly when gene essentiality predictions in the curated genome-scale model were not compromised.

## Gene essentiality analysis

To identify essential genes of Mtb grown on individual conditions (cholesterol minimal medium and glycerol minimal medium [62]), we use the Bayesian/Gumbel method of TRANSIT, version 2.02 [67]. The Bayesian/Gumbel method determines posterior probability of the essentiality of each gene (zbar). When zbar value is 1, or close to 1, the gene is considered essential (ES), if zbar is 0, or close to 0, the gene is considered non-essential (NE), uncertain (U) genes are those with zbar values between 0 and 1, and for too small (S) genes zbar is -1. After loading the TA count files (replicates for cholesterol and glycerol) and the gene annotation file into TRANSIT, and running the Gumbel method with default parameters, we obtained an output file with essentiality results (S11 and S12 Tables). Uncertain (U) and too small (S) genes were not taken into account for the *in silico* essentiality analysis. Minato and colleagues used the same statistical method to classify essential genes [65]. Conversely, DeJesus and colleagues [64] used a Hidden Markov Model based statistical method for classifying genes into four essentiality states: essential (ES), growth defect (GD), nonessential (NE), and growth advantage (GA). In order to evaluate the performance of the Mtb GSMNs to predict gene essentiality data, we used only binary classifiers, therefore we reclassify these genes just in two groups as follows: NE genes included NE and GA genes, and ES genes included GD and ES genes.

For the *in silico* gene essentiality analysis, we set the simulation conditions (asparagine, phosphate, sodium, ammonium, citrate, sulfate, zinc, calcium, chloride, $Fe^{3+}$, $Fe^{2+}$, and glycerol or cholesterol) according to Griffin minimal medium [62], 7H9 OADC medium, and "MtbYM" medium and a FBA-based gene essentiality analysis was performed in the eight Mtb models using the "single gene deletion" function of Cobra Toolbox (S8 File). Default maximization of biomass objective function was used to predict growth in all models. If a specific growth rate of no more than 5% of the wild-type was obtained, the gene was considered as essential (*in silico*), otherwise it was deemed non-essential.

Percentage of *in silico* gene essentiality predictions were categorized as: true-positive, false-positive, true-negative, and false-negative when the *in silico* data were compared with experimental essentiality data.

TP (true-positive): model simulation predicts no growth when essential genes are deleted.

FP (false-positive): model simulation predicts no growth when not essential genes are deleted.

TN (true-negative): model simulation predicts growth when not essential genes are deleted.

FN (false-negative): model simulation predicts growth when essential genes are deleted.

For evaluate the performance of the Mtb GSMNs we used sensitivity, specificity, accuracy, and Matthews Correlation Coefficient (MCC) metrics:

$$sensitivity = \frac{TP}{TP + FN} \tag{5}$$

$$specificity = \frac{TN}{TN + FP} \tag{6}$$

$$Accuracy = \frac{(TP + TN)}{(TP + FP + TN + FN)} \tag{7}$$

$$MCC = \frac{(TP * TN) - (FP * FN)}{\sqrt{(TP + FP)(TP + FN)(TN + FP)(TN + FN)}} \tag{8}$$

## Utilization of carbon and nitrogen sources

The methodology for modeling the effect of different carbon sources and nitrogen sources on Mtb growth was adapted from Lofthouse and colleagues [14]. The Biolog Phenotype MicroArray experiments classification used were obtained from Lofthouse and colleagues, 2013. They classified growth and no-growth in different carbon and nitrogen sources from the original Biolog data of Khatri and colleagues and the Roisin's minimal media [14,72]. In addition, we used Roisin's minimal media data that also were obtained by Lofthouse and colleagues.

To model the carbon source experiment, we simulated the media as a modified form of Roisin's minimal media containing unlimited quantities of ammonia, phosphate, iron, sulfate, carbon dioxide and a Biolog carbon source influx of 1 mmol/gDW/h. Similarly, the nitrogen source experiment was simulated using a modified form of Roisin's media, where ammonia was replaced with 1 mmol/gDW/h of the Biolog nitrogen source and pyruvate was used as a carbon source (influx at 1 mmol/g DW/h).

To compare the utilization of carbon and nitrogen sources in all Mtb models with experimental data, we used Matthews Correlation Coefficient metrics (Eq 8).

*In silico* growth predictions in carbon and nitrogen sources also were categorized as: true-positive, false-positive, true-negative, and false-negative.

TP (true-positive): model simulation predicts growth while growth is observed experimentally (or respiration rate is observed in Biolog phenotype microarrays) in presence of the unique carbon or nitrogen source.

FP (false-positive): model simulation predicts growth while no growth is observed experimentally in presence of the unique carbon or nitrogen source.

TN (true-negative): model simulation predicts no growth while no growth is observed experimentally in presence of the unique carbon or nitrogen source.

FN (false-negative): model simulation predicts no growth while growth is observed experimentally in presence of the unique carbon or nitrogen source.

## Pathway utilization analysis

Pathway utilization analysis differences between sMtb2.0 and iEK1011_2.0 was based on FVA and flux sampling on Roisin's media (plus glycerol and oleic acid) using the default biomass objective functions.

The FVA was run by using the function "fluxVariability" of COBRA Toolbox v.3.0 and their results were compared with the Jaccard index for each reaction in CCM and EX reactions. As suggested by Haraldsdóttir and Colleagues [106] (S4 File), the Jaccard index can be defined as the ratio between the intersection and union of the flux ranges in the sMtb2.0 and iEK1011_2.0 models (Jaccard index of 0 means disjoint flux ranges and a Jaccard index of 1 indicates completely overlapping flux ranges). The mean Jaccard index means that there is an overall similarity between flux ranges of CCM and EX reactions in both Mtb models.

The coordinate hit-and-run with rounding (CHRR) [106] algorithm was used for sampling the solution space of both Mtb models. The COBRA function "sampleCbModel" was used for running the CHRR algorithm with the following parameters: the sampling density, nStepsPerPoint = 1848 and the number of samples, nPointsReturned = 5000. A Kruskal–Wallis test (S4 File) was used to assess whether flux samples generated using either the sMtb2.0 or iEK1011_2.0 constrained with Roisin's media stemmed from the same distribution [107].

## Supporting information

**S1 Appendix. Overview of the eight constraint-based models of Mtb used in this study.** A document providing additional detail about the eight Mtb GSMNs used in this study.
(DOCX)

**S2 Appendix. Curation of additional Thermodynamic Infeasible Cycles in sMtb2.0.** A document providing detailed description of the curation of TICs in sMtb2.0.
(DOCX)

**S3 Appendix. Curation of additional Thermodynamic Infeasible Cycles in iEK1011_2.0.** A document providing detailed description of the curation of TICs in iEK1011_2.0.
(DOCX)

**S1 Fig. ROC curves of gene essentiality predictions for Mtb GSMNs. a** Receiver operating characteristic curve for the gene essentiality predictions in cholesterol minimal medium, **b** Receiver operating characteristic curve for the gene essentiality predictions in glycerol minimal medium, **c** Receiver operating characteristic curve for gene essentiality predictions in 7H9 Middlebrook OADC medium, **d** Receiver operating characteristic curve for gene essentiality predictions in MtbYM medium.
(TIF)

**S2 Fig. Central role of succinate dehydrogenase in oxidation of odd-chain substrates.**
(TIF)

**S1 Table. Set analysis of genes from all the eight Mtb GSMNs.** A table with pairwise comparisons, unions, and intersections of genes annotated in the eight GSMNs of Mtb.
(XLSX)

**S2 Table. Metabolic pathways associated to all the intersected gene sets of Mtb GSMNs.**
(XLSX)

**S3 Table. Genes and metabolic pathways shared between GSMNs of Mtb.**
(XLSX)

**S4 Table. Network topological properties of Mtb GSMNs.**
(XLSX)

**S5 Table. List of unbalanced reactions and Metabolites without formulas in the Mtb GSMNs.**
(XLSX)

**S6 Table. List of literature used in the macromolecular composition of biomass reactions of GSMN-TB and iNJ661.**
(XLSX)

**S7 Table. Biomass growth rates in complete medium.**
(XLSX)

**S8 Table. Differences in stoichiometric coefficients of BIOMASS_2 from iEK1011 compared with default biomass objective functions of sMtb and iOSDD890.**
(DOCX)

**S9 Table. List of blocked reactions and dead-end metabolites.**
(XLSX)

**S10 Table. List of Thermodynamically infeasible cycles identified for the eight Mtb GSMNs.**
(XLSX)

**S11 Table. Transposon sequencing analysis of Mtb genes required for growing on minimal medium plus cholesterol.** A list of genes of Mtb classified as essential, non-essential, and uncertain for growing in cholesterol minimal medium, the essentiality analysis was obtained by applying the Bayesian/Gumbel Method incorporated into the software TRANSIT.
(XLSX)

**S12 Table. Transposon sequencing analysis of Mtb genes required for growing on minimal medium plus glycerol.** A list of genes of Mtb classified as essential, non-essential, and uncertain for growing in glycerol minimal medium, the essentiality analysis was obtained by applying the Bayesian/Gumbel Method incorporated into the software TRANSIT.
(XLSX)

**S13 Table. Predictive power of Mtb GSMNs for classifying essential and non-essential genes on cholesterol minimal medium.** Genes whose *in silico* knockouts give growth rate values lesser than 5% of the maximum growth rate are classified as essential, otherwise are classified as non-essential.
(XLSX)

**S14 Table. Predictive power of Mtb GSMNs for classifying essential and non-essential genes on glycerol minimal medium.** Genes whose *in silico* knockouts give growth rate values lesser than 5% of the maximum growth rate are classified as essential, otherwise are classified as non-essential.
(XLSX)

**S15 Table. Predictive power of Mtb GSMNs for classifying essential and non-essential genes on 7H9 OADC medium.** Genes whose *in silico* knockouts give growth rate values lesser than 5% of the maximum growth rate are classified as essential, otherwise are classified as non-essential.
(XLSX)

**S16 Table. Predictive power of Mtb GSMNs for classifying essential and non-essential genes on YM rich medium.** *In silico* gene knockouts that have growth rates of less than 5% of the wild type growth rate are classified as essential, otherwise are classified as non-essential.
(XLSX)

**S17 Table. List of common False Positive and False Negative Genes of all the Mtb GSMNs during gene essentiality predictions.**
(XLSX)

**S18 Table. Predictive power of Mtb GSMNs growing on sole carbon sources.**
(XLSX)

**S19 Table. Predictive power of Mtb GSMNs growing on sole nitrogen sources.**
(XLSX)

**S20 Table. New added reactions into the sMtb and iEK1011 models.**
(XLSX)

**S21 Table. List of reactions with modified gene annotation in sMtb2.0.**
(XLSX)

**S22 Table. Null Space of the stoichiometric matrix formed by unbounded reactions of sMtb2.0 and iEK1011_2.0.**
(XLSX)

**S23 Table. Gibbs free energy change of Unbounded Reactions of updated Mtb GSMNs.**
(XLSX)

**S24 Table. Gene Essentiality Predictions for iEK1011_2.0 and sMtb2.0 on four Mtb media.**
(XLSX)

**S25 Table. Growth Phenotypes of iEK1011_2.0 and sMtb2.0 on unique carbon and nitrogen sources.**
(XLSX)

**S26 Table. FVA flux ranges and FBA fluxes sMtb2.0 and iEK1011_2.0 growing on Roisin's media, using the default biomass objective function as constraints.**
(XLSX)

**S27 Table. FVA and FBA fluxes of sMtb2.0 and iEK1011_2.0 growing on Roisin's medium without a defined biomass objective function.**
(XLSX)

**S1 File. Biomass growth rate comparison across Mtb GSMNs.**
(ZIP)

**S2 File. Matlab scripts for identifying unbounded reactions and thermodynamically infeasible cycles in Mtb GSMNs.**
(ZIP)

**S3 File. MEMOTE reports for iEK1011_2.0 and sMtb2.0.**
(ZIP)

**S4 File. Matlab script of FVA, FBA and Uniform Sampling for exploring solution space of iEK1011_2.0 and sMtb2.0.**
(ZIP)

**S5 File. Updated Mtb models sMtb2.0 and iEK1011_2.0 are in.mat, json, sbml, and xlsx format.**
(ZIP)

**S6 File. Matlab script for checking the charge and mass balance of Mtb GSMNs.**
(ZIP)

**S7 File. Matlab scripts for identifying Mtb GSMNs with Energy Generating Cycles.**
(ZIP)

**S8 File. Matlab scripts for gene essentiality analysis of Mtb GSMNs on four media conditions.**
(ZIP)

## Acknowledgments

Víctor A. López-Agudelo thank Msc. Laura P. Pedraza-Palacios for valuable discussions and suggestions.

## Author Contributions

**Conceptualization:** Víctor A. López-Agudelo, Dany J.V. Beste, Rigoberto Rios-Estepa.

**Data curation:** Víctor A. López-Agudelo, HuiHai Wu.

**Formal analysis:** Víctor A. López-Agudelo, Tom A. Mendum, Emma Laing, HuiHai Wu, Dany J.V. Beste, Rigoberto Rios-Estepa.

**Funding acquisition:** Dany J.V. Beste, Rigoberto Rios-Estepa.

**Investigation:** Víctor A. López-Agudelo, Tom A. Mendum, Emma Laing, HuiHai Wu, Andres Baena, Luis F. Barrera, Dany J.V. Beste, Rigoberto Rios-Estepa.

**Methodology:** Víctor A. López-Agudelo, Tom A. Mendum, Dany J.V. Beste, Rigoberto Rios-Estepa.

**Project administration:** Dany J.V. Beste, Rigoberto Rios-Estepa.

**Resources:** Víctor A. López-Agudelo, Tom A. Mendum, Emma Laing, Andres Baena, Luis F. Barrera, Dany J.V. Beste, Rigoberto Rios-Estepa.

**Software:** Víctor A. López-Agudelo, Tom A. Mendum, HuiHai Wu, Dany J.V. Beste.

**Supervision:** Dany J.V. Beste, Rigoberto Rios-Estepa.

**Validation:** Tom A. Mendum, Dany J.V. Beste, Rigoberto Rios-Estepa.

**Visualization:** Víctor A. López-Agudelo.

**Writing – original draft:** Víctor A. López-Agudelo, Andres Baena, Luis F. Barrera, Dany J.V. Beste, Rigoberto Rios-Estepa.

**Writing – review & editing:** Víctor A. López-Agudelo, Tom A. Mendum, Dany J.V. Beste, Rigoberto Rios-Estepa.

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
