## [Decision Letter · Decision Letter 0]

13 Dec 2019

Dear Dr Beste,

Thank you very much for submitting your manuscript 'A systematic evaluation of Mycobacterium tuberculosis Genome-Scale Metabolic Networks' for review by PLOS Computational Biology. Your manuscript has been fully evaluated by the PLOS Computational Biology editorial team and in this case also by independent peer reviewers. The reviewers appreciated the attention to an important problem, but raised some substantial concerns about the manuscript as it currently stands. While your manuscript cannot be accepted in its present form, we are willing to consider a revised version in which the issues raised by the reviewers have been adequately addressed. We cannot, of course, promise publication at that time.

Sincerely,

Anders Wallqvist

Associate Editor

PLOS Computational Biology

William Noble

Deputy Editor

PLOS Computational Biology

[LINK]

Reviewer's Responses to Questions

**Comments to the Authors:**

Reviewer #1: In this paper, the authors evaluate 8 published genome-scale models (GEMs) of M. tuberculosis H3Rv and refine the best performing GEMs for future use. Specifically, the authors compare the gene content, currency metabolite connectivity, biomass reactions, charge balance, mass balance, blocked reactions, thermodynamics, gene essentiality predictions, and utilization of carbon sources across the different models. The authors use standard metrics to assess the various model performances and also place the model predictions within the context of TB literature to help discern biological relevance. The authors identify iEK1011 and sMtb2018 as the best overall GEM and improve their quality by adding various metabolic reactions to the models with detailed reasons. Given the power of GEMs to enable deep analysis of TB metabolism, as well as the large availability and variability in TB GEMs, we believe that this study adds significant value to the TB research community and thus should be published in PLOS Computational Biology. However, there are more GEM evaluations and details described below in the major comments that we would like to see the authors address before publication. In particular, we would like to see the authors detail differences in pathway utilization strategies between the improved models, if there are any.

To the Authors:

Major comments

1. It is highly recommended by the COBRA community that the authors provide a MEMOTE report of both the compared and improved reconstructions (https://www.biorxiv.org/content/10.1101/350991v1). Rather than dictating whether the models are worthy of being published, the MEMOTE report should help understand what the limitations are and where they should focus their efforts in potentially improving the model. Importantly, the MEMOTE report may help the authors improve improve the refined models and comparisons before publication.

2. The authors should additionally provide the improved reconstructions in both json and sbml format.

The authors should add another section detailing the different biomass reactions across the models (i.e., metabolite composition, metabolite weightings, shadow prices, etc). Currently, only a comparison of the biomass molecular weight is provided at the end of the “Checking mass and charge balances of biochemical reactions” section.

3. The authors should provide a description/comparison of the growth-associated maintenance (GAM) or non-growth associated maintenance (NGAM) reactions across the different TB models. One or two sentences is sufficient.

4. Both flux variability analysis (FVA) and flux sampling are key FBA tools used to evaluate the solution space of GEMs. We recommend that the authors perform FVA and flux sampling of the improved GEMs (iEK1011_2.0 and sMtb2.0) to both evaluate the flux bottlenecks and distributions between models.

5. Continuing from comment 5, the authors should provide detailed comparisons of the flux profile between the improved GEMs (iEK1011_2.0 and sMtb2.0) in standard TB media using either parsimonious FBA (pFBA), FVA, or flux sampling. For example, do the models prefer different nutrients? Do they have similar flux strategies in TCA and oxPPP? We would like to see the authors detail differences in pathway utilization strategies between the improved models, if there are any.

Minor comments

1. Line 100: The description of red in Fig 2 is written the same as blue.

Reviewer #2: In their manuscript titled “A systematic evaluation of Mycobacterium tuberculosis Genome-Scale Metabolic Networks” Lope-agudelo et al. have systematically examined 8 of the most recent Mtb GEM models and have identified a significant number of problems with them. As I note below, I have questions about some of their choices but overall, I congratulate and thank the authors for a detailed examination of models for an organism of significant importance.

Despite this, I don’t find the work novel and suited for Plos Computational biology. I expect Plos Comp bio papers to either introduce a novel computational biology method or report a significant biological find. Neither of these apply to this paper. The manuscript is better suited for more general journals like Plos One or Scientific reports.

Perhaps if the authors had generated a single unified improved model of Tb metabolism, then it would have been appropriate for Plos Comp Bio. As it stands, by their own admission, one of their models is better suited for interacting with Recon model while the other is better suited for simulating growth under a variety of different conditions.

Major concerns

• The paragraph starting on line 38 is not sufficient to describe the examined models and how they differ. The order of publication and combination of models is included in figure 1 and further detail is provided in S1 but there is no mention of it in the main text.

• The model with the greatest number of reactions and genes is iAB-AMF-1410-Mt-661 updated published on 2017. Why was this model ignored? There is no mention of this model. How can the purported updated models be considered most complete if they still have fewer reactions and genes than the noted model?

• Examining figure 2 it is obvious that some models are just the old models plus some new reactions e.g. sMtb and sMtb2018. Why include the former when instead iAB-AMF-1410-Mt-661 could have been examined?

• Line 63, what do you mean by multi-scale simulation platform? What scales are you talking about?

• Line 92, I’m confused about the statement “has poor annotation of genes”? Does that mean the genes identified were incorrectly assigned a function or that the coverage of the pathways was lacking?

• How does iCG760 operate as a system-level model without water as one of its metabolites? To me the sorry biochemical states of three of the published models is the most shocking find of the paper.

• When highlighting the results of analyses for blocked reactions and dead-end metabolites it is important to note if the cause is due to gaps in genome or incomplete GPRs. The former could be viewed as the modeler trying not to add a new phenotype without experimental validation whereas the latter is a poorly curated model.

• Why not combine sMtb2.0 and iEK1011_2.0 into one unified model that would work well with Recon for host pathogen interactions as well as correctly predicts growth on a variety of different compounds? That would be the perfect outcome from your extensive efforts.

Minor concerns

• Do not use acronyms in the abstract prior to defining them, i.e. TB

• Some figures are labeled Fig. others Figure. Please choose one format.

• The sentence starting on line 152 is incomplete.

Reviewer #3: The article entitled, 'A systematic evaluation of Mycobacterium tuberculosis Genome-Scale Metabolic Networks', Lopez-Agudelo et al present a comparative analysis of Mtb genome scale metabolic reconstructions.

This is a welcome analysis in the literature, given the growing number of models and derivations of models and the need for reconciliation and assessment of the current state. While the most recent update iEK1011 presented an update, the authors identify aspects of the model that area missing.

+ Figure 1: Provides a nice summary of an otherwise confusing development of multiple models using genome-scale models of Mtb.

+ p7: How did the authors arrive at the cutoff of 10% for the deviation of 1 g/mmol? Why is iEK1011 so much higher than some of the other models?

+ Fig 5: It is understandable to revise reaction directionality based on thermodynamic feasibility assessment, however to remove a particular reaction(s) (DFRA1 and DFRA3) should require a little more justification, including any experimental support/arguments and/or explanations of why the arguments made for a different genus (yeast) are also relevant for Mtb.

+ p13: the identification of a need for a menaquinone-dependent succinate dehydrogenase reaction is an interesting modification that appeared to improve the predictive capability of the model. However the authors state that they had to add this reaction to iEK1011, whereas in reality all that was required was to change this from an irreversible to a reversible reaction. It would be helpful to note whether there is experimental evidence to suggest that concentrations are present to enable thermodynamic reversibility of the reaction. Additionally, will the reversible form of the reaction result in any infeasible reaction cycles?

+ p16: similar to the question raised on p13, which one of these involved addition of new reactions vs making reactions reversible

+ p18: 'thermodynamically unstable' -- perhaps the authors meant thermodynamically infeasible, since there does not appear to be any discussion of stability of particular flux states

+ p17: what specifically is meant by "unblocking B12 synthesis"? The authors themselves cite specific articles that highlight the incomplete knowledge about B12 handling in Mtb.

+ p17: it is not clear what addition is made by addition of biotin?

+ p17: again, it is not clear what specifically the authors mean by "including ... pyridoxal-5-phosphate", although from the referenced article, in this case they presumably are referring to the addition of pyridoxal 5' synthase. For biotin, it is not clear what revisions they made or the rational/justification for it.

+ Table 5: it is not clear where the cited values for O2, NADH/NAD, and NADPH/NADP are obtained from the referenced citations, since the paper and textbook (non-specific reference) do not appear to cite measurements for Mtb. Additionally it is not clear what is meant by "Other metabolites".

+ p26: What was the scientific rationale/calculation to identify the 5% cutoff for

+ minor comments:

there are a number of small grammatical errors, e.g.

p13, l255: "is as a result"

p16, l312: "exempler", did you mean exemplar?

p 29: supplemental file legends: S10, S11, S12, S13, "for classify essential"

etc.

**Have all data underlying the figures and results presented in the manuscript been provided?**

Reviewer #1: Yes

Reviewer #2: Yes

Reviewer #3: Yes

PLOS authors have the option to publish the peer review history of their article (what does this mean?). If published, this will include your full peer review and any attached files.

Reviewer #1: No

Reviewer #2: No

Reviewer #3: No

---

## [Decision Letter · Decision Letter 1]

27 Apr 2020

Dear Dr. Beste,

Thank you very much for submitting your manuscript "A systematic evaluation of Mycobacterium tuberculosis Genome-Scale Metabolic Networks" for consideration at PLOS Computational Biology. As with all papers reviewed by the journal, your manuscript was reviewed by members of the editorial board and by several independent reviewers. The reviewers appreciated the attention to an important topic. Based on the reviews, we are likely to accept this manuscript for publication, providing that you modify the manuscript according to the review recommendations.

Sincerely,

Anders Wallqvist

Associate Editor

PLOS Computational Biology

William Noble

Deputy Editor

PLOS Computational Biology

[LINK]

Reviewer's Responses to Questions

**Comments to the Authors:**

Reviewer #1: We are happy with the authors revisions and have no further comments.

Reviewer #2: The authors have answered all my concerns.

Reviewer #3: The revised version provides clarifications for many of the points that were raised. Further the incorporation of MEMOTE report, as suggested by a different reviewer, is a nice addition.

A few follow up points are worth addressing,

+ Regarding the question of Vitamin B12 synthesis, regardless of whether another paper has included the pathway based on ‘physiological’ argument is generally not considered sufficient criteria for incorporating content into genome-scale metabolic networks (especially if the purpose of the manuscript is to provide clarity in a field with multiple versions of models). Generally even if ‘physiological’ evidence arguments are made, this typically involves inclusion of a single enzyme or transporter, thus the inclusion of multiple enzymatic steps introduces the potential for significant error into the model.

The manuscript by Minas et al poses a possible pathway, but lacks any strong biochemical evidence for the actual enzymes and pathway. The authors should clearly identify which enzymes/reactions were added. Microbial vitamin B12 synthesis is not a trivial or singular pathway, as recently summarized by Fang et al, Microbial production of vitamin B12: a review and future perspectives, DOI 10.1186/s12934-017-0631-y. It should clearly be stated which enzymes/reactions were added and ideally some type of justification (it is recognized that in the absence of direct evidence

+ “We also included the co-factors biotin and pyridoxal-5-phosphate in the biomass formulation to enhance the phenotype prediction of sMtb2018 and iEK1011 as recommended by Xavier et al. [19].”

The argument that the authors defer to Xavier et al regarding inclusion of pyridoxa-5-phosphate into the model, is not the standard that genome-scale reconstructions are typically constructed with. However by provided two versions of the biomass function (“BiomassGrowth_2.0” and “BIOMASS__2.1”), this issue is adequately addressed.

+ There are still a few grammatical/spelling errors, e.g. Mathews Correlation Coefficient should read Matthews Correlation Coefficient, the authors likely mean ‘bona fide’ and not ‘bone-fide’ on page 10, etc. Recommend one more read through by authors to minimize misspellings.

**Have all data underlying the figures and results presented in the manuscript been provided?**

Reviewer #1: Yes

Reviewer #2: Yes

Reviewer #3: Yes

PLOS authors have the option to publish the peer review history of their article (what does this mean?). If published, this will include your full peer review and any attached files.

Reviewer #1: No

Reviewer #2: No

Reviewer #3: No
---

## [Editor Report · Decision Letter 2]

8 May 2020

Dear Dr. Beste,

We are pleased to inform you that your manuscript 'A systematic evaluation of Mycobacterium tuberculosis Genome-Scale Metabolic Networks' has been provisionally accepted for publication in PLOS Computational Biology.

Best regards,

Anders Wallqvist

Associate Editor

PLOS Computational Biology

William Noble

Deputy Editor

PLOS Computational Biology

---

## [Editor Report · Acceptance letter]

5 Jun 2020

PCOMPBIOL-D-19-01934R2 

A systematic evaluation of Mycobacterium tuberculosis Genome-Scale Metabolic Networks

Dear Dr Beste,

I am pleased to inform you that your manuscript has been formally accepted for publication in PLOS Computational Biology. Your manuscript is now with our production department and you will be notified of the publication date in due course.

With kind regards,

Sarah Hammond
